# Drosulfakinin signaling in *fruitless* circuitry antagonizes P1 neurons to regulate sexual arousal in *Drosophila*

Shunfan Wu[1,2,8], Chao Guo [1,8], Huan Zhao[1,8], Mengshi Sun[1], Jie Chen[1], Caihong Han[1], Qionglin Peng[1], Huanhuan Qiao[3,4], Ping Peng[3,4], Yan Liu [5], Shengzhan D. Luo[6] & Yufeng Pan [1,7]*

Animals perform or terminate particular behaviors by integrating external cues and internal states through neural circuits. Identifying neural substrates and their molecular modulators promoting or inhibiting animal behaviors are key steps to understand how neural circuits control behaviors. Here, we identify the Cholecystokinin-like peptide Drosulfakinin (DSK) that functions at single-neuron resolution to suppress male sexual behavior in *Drosophila*. We found that *Dsk* neurons physiologically interact with male-specific P1 neurons, part of a command center for male sexual behaviors, and function oppositely to regulate multiple arousal-related behaviors including sex, sleep and spontaneous walking. We further found that the DSK-2 peptide functions through its receptor CCKLR-17D3 to suppress sexual behaviors in flies. Such a neuropeptide circuit largely overlaps with the *fruitless*-expressing neural circuit that governs most aspects of male sexual behaviors. Thus DSK/CCKLR signaling in the sex circuitry functions antagonistically with P1 neurons to balance arousal levels and modulate sexual behaviors.

[1] The Key Laboratory of Developmental Genes and Human Disease, Institute of Life Sciences, Southeast University, Nanjing 210096, China. [2] College of Plant Protection, Nanjing Agricultural University, Nanjing 210095, China. [3] School of Medicine, Tsinghua University, Beijing 100084, China. [4] Tsinghua Fly Center, Tsinghua University, Beijing 100084, China. [5] Institute for Stem Cell and Neural Regeneration, School of Pharmacy, Nanjing Medical University, Nanjing 211166, China. [6] Janelia Research Campus, Howard Hughes Medical Institute, 19700 Helix Drive, Ashburn, VA 20147, USA. [7] Co-innovation Center of Neuroregeneration, Nantong University, Nantong 226019, China. [8] These authors contributed equally: Shunfan Wu, Chao Guo, Huan Zhao. *email: pany@seu.edu.cn

Animal behaviors are modulated by both internal states and external stimuli. When an organism is faced with more than one stimulus in the context of distinct behavioral states, multiple decision-making processes are involved in making appropriate behavioral choices. A fundamental question in neuroscience is to understand how neural circuits and molecular modulators control these decision-making processes. Significant progress has been made in identifying the molecules and neurons that control innate and social behaviors, such as courtship, sleep, feeding, and aggression in *Drosophila*[1–5] and mouse[6–9].

Male courtship in *Drosophila melanogaster* is one of the best-understood innate behaviors, and largely controlled by the *fruitless* (*fru*) gene and *doublesex* (*dsx*) gene, which encode sex-specific transcription factors (FRU$^M$ and DSX$^M$ in males and DSX$^F$ in females)[10–12]. FRU$^M$ is responsible for most aspects of male courtship[13–15], and DSX$^M$ is important for the experience-dependent acquisition of courtship in the absence of FRU$^M$, and courtship intensity and sine song production in the presence of FRU$^M$ [16,17]. FRU$^M$ is expressed in a dispersed subset of ca. 2000 neurons including sensory neurons, interneurons, and motor neurons that are potentially interconnected to form a sex circuitry controlling sexual behaviors[14,15,18,19]. In contrast, DSX$^M$ is expressed in ca. 700 neurons in males, the majority of which also express FRU$^M$, and are crucial for male courtship[20,21]. Recently, substantial progress has been made into how external sensory cues are perceived and integrated by *fru$^M$* and/or *dsx$^M$* neurons to initiate male courtship, in particular, how a subset of male-specific *fru$^M$*- and *dsx$^M$*-expressing P1 neurons integrate olfactory and gustatory cues from female or male targets to initiate or terminate courtship[22–26]. Such a neuronal pathway is also conserved in other *Drosophila* species[27–29].

Behavioral decisions depend on both excitatory and inhibitory modulations. P1 neurons represent an excitatory center that integrates multiple (both excitatory and inhibitory) sensory cues and initiates courtship[4,24,25]. However, whether there is an inhibitory counterpart that operates against P1 neurons to balance sexual activity is still unknown. Indeed, males do not absolutely court virgin females even if these females may provide the same visual, olfactory, and gustatory cues, depending on the male's internal states and past experiences. It has been previously shown that neuropeptide SIFamide acts on *fru$^M$*-positive neurons and inhibits male–male but not male–female courtship[30,31], and SIFamide neurons also integrate multiple peptidergic neurons to orchestrate feeding behaviors[32], but whether SIFamide inhibits internal arousal states for sexual behaviors is not clear. Recently, we found that sleep and sex circuitries interact mutually and demonstrate how DN1 neurons in the sleep circuitry and P1 neurons in the courtship circuitry function together to coordinate behavioral choices between sleep and sex[33]. However, we know very little on the inhibitory pathway(s) that may represent internal arousal states and inhibit courtship toward females.

In this study, we set out to identify courtship inhibitory neurons that express neuropeptides in *Drosophila*, as neuropeptides play key roles in adjusting animal behaviors based on environmental cues and internal needs[34,35]. We identify the neuropeptide Drosulfakinin (DSK), the fly ortholog of Cholecystokinin (CCK) in mammals, functions through its receptor CCKLR-17D3 in the *fru$^M$*-expressing sex circuitry to inhibit male courtship toward females. We further demonstrate that *Dsk* neurons and P1 neurons interact and oppositely regulate male sexual behaviors.

## Results

### *Drosulfakinin-GAL4* neurons inhibit male courtship behavior.
We speculated that neuropeptides might function as molecular modulators in courtship circuit to control courtship behaviors,

and screened for courtship deficit using 32 *GAL4* lines driving the temperature-sensitive activator dTrpA1[36] in distinct subsets of peptidergic neurons[37,38] (Fig. 1a, b).

This screen identified three *GAL4* lines when combined with *UAS-dTrpA1* at 30 °C severely impaired male courtship, including two Corazonin (*Crz*) *GAL4* lines and a *Drosulfakinin* (*Dsk*) *GAL4* line (courtship index [CI] < 20%, which is the fraction of observation time that males courted, Fig. 1b). Further analysis revealed that activating *CrzGAL4* neurons[37], but not *DskGAL4* neurons, induced rapid ejaculation in isolated males (Supplementary Table 1). We also identified a *Myoinhibitory-peptide* (*Mip*) *GAL4* line when combined with *UAS-dTrpA1* at 30 °C mildly inhibited male courtship (CI~54%), but such inhibition was not consistent using two other *Mip-GAL4* drivers (CIs > 80%, Fig. 1b). We also found that activation of neurons labeled by two *SIFaGAL4* drivers did not affect male courtship toward females (CIs > 80%, Fig. 1b), although it was previously shown that SIFamide neurons inhibit male–male courtship[30,31]. Thus, we focused our further study on *Dsk* and *Dsk*-expressing neurons.

The above P-element based *DskGAL4* labels a subset of *Dsk*-expressing neurons as well as a few non-*Dsk* neurons as revealed by GFP and anti-DSK (Supplementary Fig. 1). To further study the function of *Dsk*-expressing neurons, we generated the PhiC31-based site-specific *DskGAL4*, *DskLexA*, and *DskFlp*, using the 1.1 kb sequence upstream of the transcription start site of the *Dsk* gene as promoter. The new *DskGAL4* specifically labels four pairs of neurons in the medial protocerebrum, and weakly labels a few insulin-producing cells (IPCs) in the Pars Intercerebralis (PI) region (Fig. 2a and Supplementary Fig. 2a), confirming a previous study[39]. Thus we used the new *DskGAL4* hereafter. Activation of these *DskGAL4*-labeled neurons via *dTrpA1* severely impaired male courtship (CI~10%, Fig. 1c, Supplementary Movie 1), while activating the *Dsk*-positive IPCs using *dilp2GAL4*[40] did not affect courtship (CI > 80%, Supplementary Fig. 2). Furthermore, activating *DskGAL4* neurons does not affect feeding, probably due to the weak labeling of IPCs (Supplementary Fig. 3). Silencing *DskGAL4* neurons does not enhance the already high level of male courtship (CIs > 80%, Supplementary Fig. 4).

To further confirm our findings, we used the optogenetic effector CsChrimson[41], which affords greater control over the dynamic range of neuronal activation than *dTrpA1*[42], and found that optogenetic stimulation of *Dsk* neurons in *UAS-CsChrimson-mVenus/DskGAL4* males almost abolished courtship such that 90% of males do not initiate courtship (CI~2%), dramatically different from all control males (CIs > 50%, Fig. 1d). Note that the empty *GAL4* control flies also showed reduced courtship under red light, which may be due to genetic background and/or red light perturbation, and we used other control lines (e.g. *DskGAL4/+*) in our later experiments. We next assayed courtship in dark condition for 2 min to allow courtship (CI$_{1-2}$ ~ 40%), and then turned on red light for optogenetic activation. We found that activation of *DskGAL4* neurons immediately abolished courtship by males that already initiated courtship (Fig. 1e). Furthermore, we tested how much time such inhibition would last by assaying courtship in dark after 10 min in red light, and found that courtship was only partially restored after ~20 min (CI$_{1-10}$ is 1%, CI$_{11-20}$ is 4.8% and CI$_{21-30}$ is 43.6%, Fig. 1f). These results indicate that *DskGAL4* neurons rapidly and efficiently inhibit male courtship and such inhibition lasts for a few minutes.

### Four pairs of *Dsk* and *fru$^M$* neurons inhibit male courtship.
Analysis of the new *DskGAL4* and *DskLexA* using a *myr::GFP* reporter revealed similar expression patterns, both of which label a subset of neurons in the brain that were previously identified as *Dsk*-expressing neurons[39], including four pairs of neurons in the

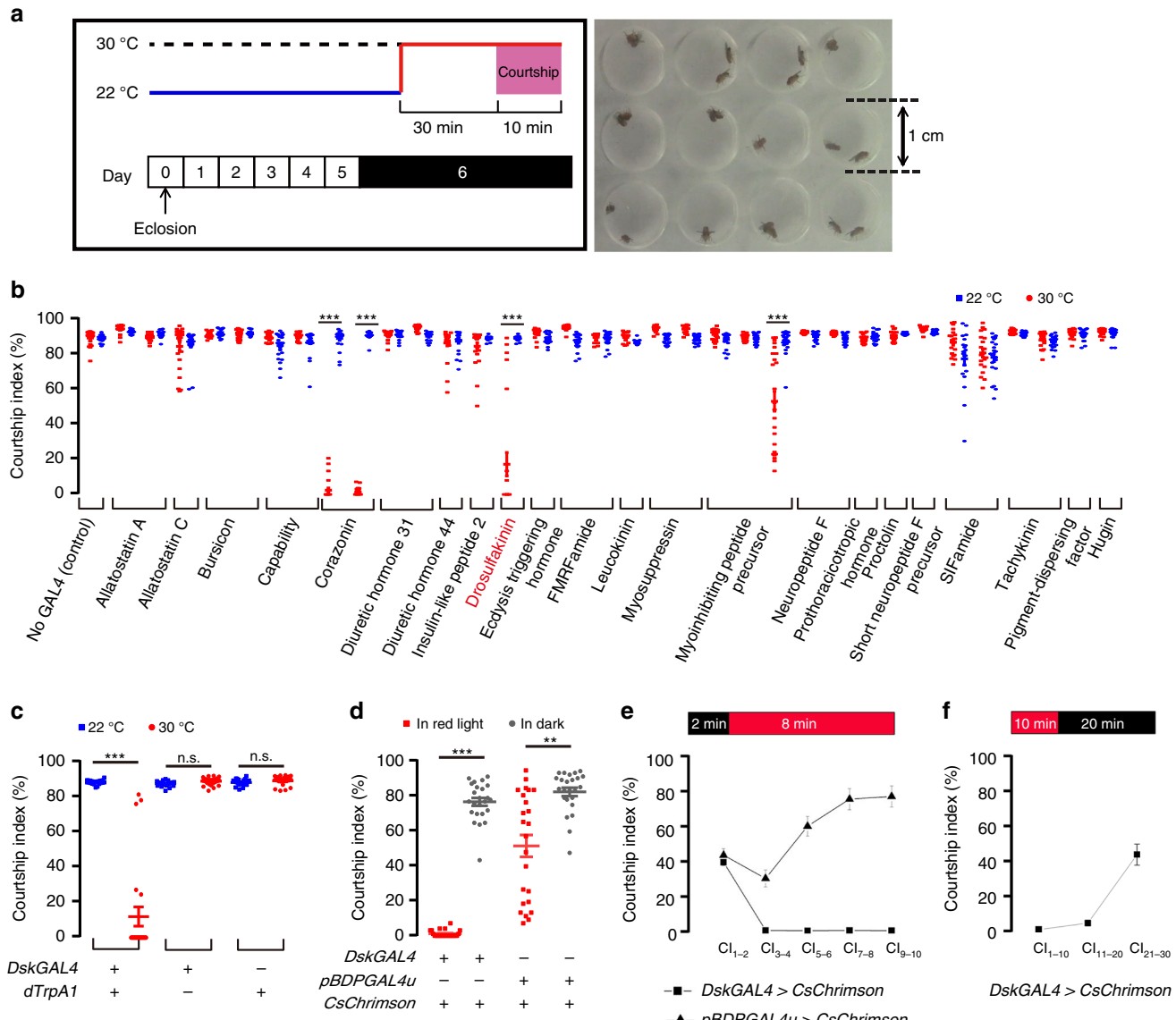

**Fig. 1** Identification of *DskGAL4* neurons that inhibit male courtship. **a** Experimental design of screening for courtship-inhibiting neurons. **b** Identification of *DskGAL4* and *CrzGAL4* that inhibit male courtship when driving *UAS-dTrpA1* at 30 °C (red circle) compared to permissive temperature 22 °C (blue square). *n* = 12 for *sNPF-GAL4* and *Lk-GAL4*, *n* = 18 for *Burs-GAL4*, *Dh31-GAL4* and *Proc-GAL4*, *n* = 20 for *SIFaGAL4s*, *n* = 24 for others. \*\*\*p < 0.001, Mann–Whitney U test. **c** Thermogenetic activation of *Dsk* neurons labeled by another *DskGAL4* (attP2) severely inhibits male courtship. *n* = 24 for each. *p* < 0.001, Kruskal–Wallis test. \*\*\*p < 0.001 and *p* > 0.1 (n.s.), post hoc Dunn's multiple comparisons test. **d** Optogenetic activation of *DskGAL4* (attP2) neurons abolishes male courtship. *n* = 24 for each. *p* < 0.001, Kruskal–Wallis test. \*\*p < 0.01 and \*\*\*p < 0.001, post hoc Dunn's multiple comparisons test. **e** Optogenetic activation of *DskGAL4* neurons rapidly inhibits male courtship. *n* = 24 for each. **f** Courtship inhibition by optogenetic activation of *DskGAL4* neurons lasts for more than 10 min after lights off. *n* = 24. n.s. not significant. Error bars indicate SEM. Source data are provided as a Source Data file

medial protocerebrum (two pairs of MP1 and two pairs of MP3 cells), and a few IPCs (Fig. 2a and Supplementary Fig. 2a). To further confirm these findings, we generated a polyclonal anti-body raised against DSK (antigen: FDDYGHMRF, recognized two DSK peptides, DSK-1 and DSK-2), which faithfully labels MP1 and MP3 cells, and a number of interneurons in both the brain and the ventral nerve cord (Fig. 2a), consistent with previous findings[39]. It should be mentioned that our *DskGAL4* did not label neurons in the VNC. It is possible that those antibody-labeled neurons in the VNC are not *Dsk*-expressing neurons, but the DSK antibody cross-reacts with other peptides in those neurons (e.g. FMRFamide-related peptides). Alternatively, those neurons in the VNC are indeed *Dsk*-positive, but the *DskGAL4* only labels a part of *Dsk*-expressing neurons. We observed no gross anatomy difference of *Dsk* neurons using both anti-GFP

and anti-DSK between males and females (Supplementary Fig. 5a), indicating that these *Dsk* neurons are not sex-specific. Thus, we also activated *DskGAL4* neurons using *dTrpA1* (Supplementary Fig. 5b) or *CsChrimson* (Supplementary Fig. 5c) in females and observed significantly reduced receptivity to courting wild-type males. These results indicate that *Dsk*-expressing neurons are common in both sexes and suppress male and female sexual behaviors, and we focused on male courtship in this study (but see below for their interaction with sex-specific neurons in both males and females).

Male courtship behavior is governed by neural circuitry comprised of ~2000 *fru*[M]-expressing neurons[14,15,18,19]. We asked if the courtship-inhibition neurons labeled by the *DskGAL4* are a part of the *fru*[M] circuitry. Double labeling of *fru*[LexA] and *DskGAL4* neurons (*LexAop-RedStinger/UAS-stinger-GFP;*

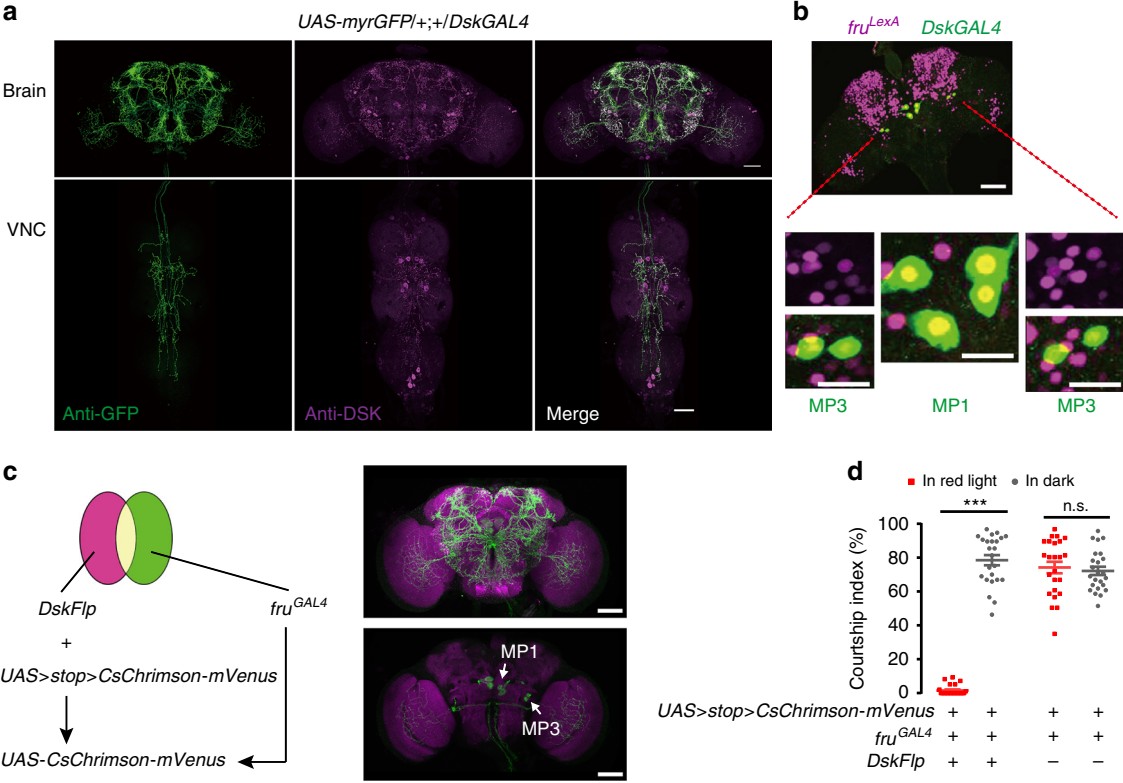

**Fig. 2** Four pairs of *Dsk*- and *fru^M*-expressing neurons inhibit male courtship. **a** Expression pattern of *DskGAL4* in the central nervous system revealed by anti-GFP (left) and anti-DSK (middle). Representative of eight male flies. Scale bars, 50 µm. **b** Two pairs of MP1 and two pairs of MP3 neurons are co-labeled by *fru^LexA* driving *LexAop-RedStinger* (magenta) and *DskGAL4* driving *UAS-Stinger-GFP* (green). Representative of five male brains. Scale bars, 50 µm and 20 µm (zoom-in). **c** Intersectional strategy to label and manipulate *Dsk* and *fru^M* co-expressing MP1 and MP3 neurons. Representative of 6 male brains. Scale bars, 50 µm. **d** Optogenetic activation of intersectional neurons between *DskFlp* and *fru^GAL4* abolishes male courtship. Red square indicates test in red light, and gray circle indicates test in dark. $n = 24$ for each. $p < 0.001$, Kruskal–Wallis test. ***$p < 0.001$ and $p > 0.99$ (n.s.), post hoc Dunn's multiple comparisons test. Error bars indicate SEM. Source data are provided as a Source Data file

*fru^LexA*/*DskGAL4*) revealed that the four pairs of MP1 and MP3 neurons are all *fru^M*-positive (Fig. 2b). We then used an intersectional strategy[23] to visualize overlapped expression between *fru^LexA* and *DskGAL4* (*UAS > stop > myrGFP/+; LexAop2-FlpL fru^LexA/DskGAL4*), but only observed stochastic MP1 and MP3 cells. However, we observed consistent expression of MP1 and MP3 cells, as well as a number of IPCs, from intersection of *fru^GAL4* and *DskFlp* (Fig. 2c). Further, we activated these overlapped neurons using the above optogenetic effector *CsChrimson*, and found that these males (*UAS > stop > CsChrimson-mVenus/+; fru^GAL4/DskFlp*) almost do not court (CI ~ 2%) under red light stimulation, while all control males court intensively (CIs > 70%, Fig. 2d). Together with the finding that IPCs are not involved in courtship suppression, these results demonstrate that two pairs of MP1 and two pairs of MP3 cells are *Dsk*- and *fru^M*-positive and responsible for courtship inhibition.

**Single-neuron labeling of the *Dsk*-expressing MP neurons**. The four pairs of *Dsk*- and *fru^M*-expressing MP neurons project to various regions of the brain, but the morphology and function of individual neurons or neuronal types are unclear. Thus we used heat-shock controlled flipase to stochastically label and activate subsets of *DskGAL4* neurons in males (*hs-Flp/Y; UAS > stop > CsChrimson-mVenus/+; DskGAL4/+*). These experiments revealed fine structures of individual *Dsk* neurons. The two pairs of MP3 cells are indistinguishable and mainly project to the superior medial protocerebrum (SMP), the superior lateral protocerebrum (SLP) and the lateral horn (LH)[43] (Fig. 3a labels a single MP3 neuron, and Fig. 3b labels two MP3 neurons). The

two pairs of MP1 cells are distinct and termed as MP1a (Figs 3c, d label single MP1a neurons) and MP1b (Fig. 3e labels a single MP1b neuron) hereafter. The MP1a and MP1b cells project to various regions of the brain including the optic lobes (specifically from MP1a), suboesophageal ganglion (SOG) and the lateral protocerebral complex where the male-specific P1 neurons integrate multiple sensory cues and initiate courtship[18,19].

Among the 103 males we assayed for courtship and later successfully imaged for *CsChrimson-mVenus* expression, 30 actively courted and 73 did not initiate courtship (for details, see methods). The frequency of *CsChrimson-mVenus* expression in MP1a and MP1b neurons, but not MP3, is significantly higher in non-courters than that in courters (76.7% of non-courters vs. 70% of courters label the MP3 cell [$p > 0.05$]; 87.7% of non-courters vs. 63.3% of courters label the MP1a cell [$p < 0.01$]; 79.5% of non-courters vs. 13.3% of courters label the MP1b cell [$p < 0.001$], Fig. 3b). Indeed, MP1a and MP1b neurons, but not MP3 neurons, project to the lateral protocerebral complex that is important for courtship initiation. Together these results illustrate the identity of individual MP cells, demonstrate their function in courtship inhibition, and suggest that the MP1a and MP1b cells may play more important roles then MP3 cells do in courtship inhibition.

**Dsk and P1 neurons antagonistically regulate sexual arousal**. It has been well established that the male-specific P1 neurons integrate chemosensory cues from potential mates and positively control male sexual arousal levels[6,23–25,42]. As *Dsk* neurons also project to the lateral protocerebral complex where inputs and

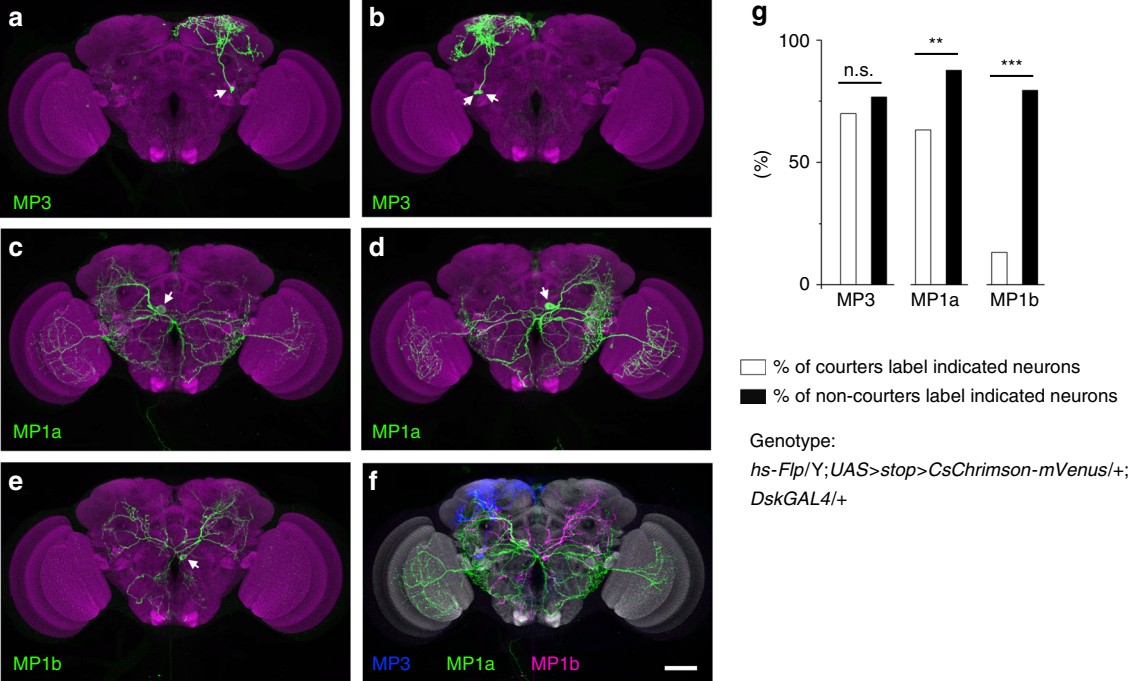

**Fig. 3** Single-neuron labeling and manipulation of *Dsk* MP neurons. **a–f** Stochastic labeling of single MP3 (**a**), two MP3 (**b**), single MP1a (**c** and **d**) and single MP1b (**e**) neurons. Two MP3, one MP1a and one MP1b neurons are registered in a standard brain (**f**). Scale bars, 50 μm. **g** The frequency of *hs-Flp/Y; UAS > stop > CsChrimson-mVenus/+; DskGAL4/+* males with *CsChrimson-mVenus* expressed in indicated neurons for courters ($n = 30$, white bars) or non-courters ($n = 73$, black bars). **$p < 0.01$, ***$p < 0.001$ and $p = 0.48$ (n.s.), chi-square test. n.s. not significant. Source data are provided as a Source Data file

outputs of P1 neurons both reside, we hypothesize that *Dsk* neurons may interact with P1 neurons to co-regulate sex and other arousal-related behaviors.

Firstly, we co-activated *Dsk* neurons and P1 neurons to test whether they function in a linear pathway where the downstream one may dominantly affect courtship. We used three independent P1 drivers (*R15A01GAL4*, *R71G01GAL4*, or *P1-splitGAL4*)[23,33,42], and in all cases, we found that co-activation of *Dsk* and P1 neurons resulted in intermediate courtship levels compared to *Dsk* activation and P1 activation alone (Fig. 4a). These results indicate that *Dsk* and P1 neurons are not in a linear pathway, and they function antagonistically to modulate male courtship behavior.

Secondly, we tested how *Dsk* and P1 neurons may affect other arousal-related behaviors, e.g., sleep. We found that activation of P1 neurons dramatically decreased male sleep as previously reported[33,44], but activation of *Dsk* neurons did not significantly affect sleep. However, we found that co-activation of *Dsk* and P1 neurons also resulted in intermediate sleep amounts (Fig. 4b, c), indicating that *Dsk* neurons indeed function antagonistically with P1 neurons on the control of sleep, in addition to the control of sexual behaviors.

To further confirm these findings, we used video tracking on individual males to measure spontaneous walking activity for 24 h, and observed similar results. Activation of P1 neurons alone promotes persistently high level of spontaneous walking for 24 h, while activation of *Dsk* neurons alone only mildly decreases spontaneous walking in the first hour, but does not significantly affect the average walking velocity for 24 h (Fig. 4f–j). Interestingly, we observed no difference between P1 activation and P1&*Dsk* co-activation males on their spontaneous walking activity during the first hour, but dramatic differences as activation proceeds, such that walking activity by P1&*Dsk* co-activation males is ~50% lower than P1 activated males, although still much higher than control males (Fig. 4f–j). These results

clearly show that *Dsk* neurons function oppositely and persistently against P1 neurons on the control of spontaneous walking activity.

That activation of *Dsk* neurons mildly decreased spontaneous walking activity (at least initially) raises the possibility of a locomotion deficit. To rule out such possibility, we assayed their locomotion providing external stimuli. First, males with activated *Dsk* neurons walk and jump as quickly as control males when mechanically perturbed (Movie S2); second, males with activated *Dsk* neurons track rotating visual stimulus normally and walk as fast as control males in a visual-induced locomotion assay (Fig. 4k–m). That *Dsk* activated males rarely court females but normally track rotating visual stimulus further indicating that *Dsk* neurons negatively regulate an internal arousal state for sex and spontaneous walking, in opposite to the function of P1 neurons.

**Dsk neurons receive synaptic transmission from P1 neurons.** The above results demonstrate that *Dsk* and P1 neurons function oppositely to regulate internal arousal levels, but whether they interact with each other is unclear. We registered individual MP3, MP1a, or MP1b neurons (Fig. 3a) with P1 neurons (labeled by *P1-splitGAL4*), and found that these MP neurons all have close contact with the P1 neurons (Fig. 5a and Supplementary Movie 3–5). Indeed, we observed substantial GRASP (GFP reconstitution across synaptic partners)[45] signal between *Dsk* and P1 neurons (Fig. 5b), suggesting these neurons might have direct synaptic connection. We then used the recently modified activity-dependent GFP reconstitution method[46]. This method uses GFP1–10 fragment fused to the presynaptic synaptobrevin (*syb::spGFP1–10*) and a membrane tethered version of the complementing split GFP (*CD4::spGFP11*), and GFP reconstitution is achieved only if there are vesicle fusion between neurons labeled by corresponding *GAL4* and *LexA* drivers. We first tested if *Dsk* neurons are presynaptic to P1 neurons by driving expression of

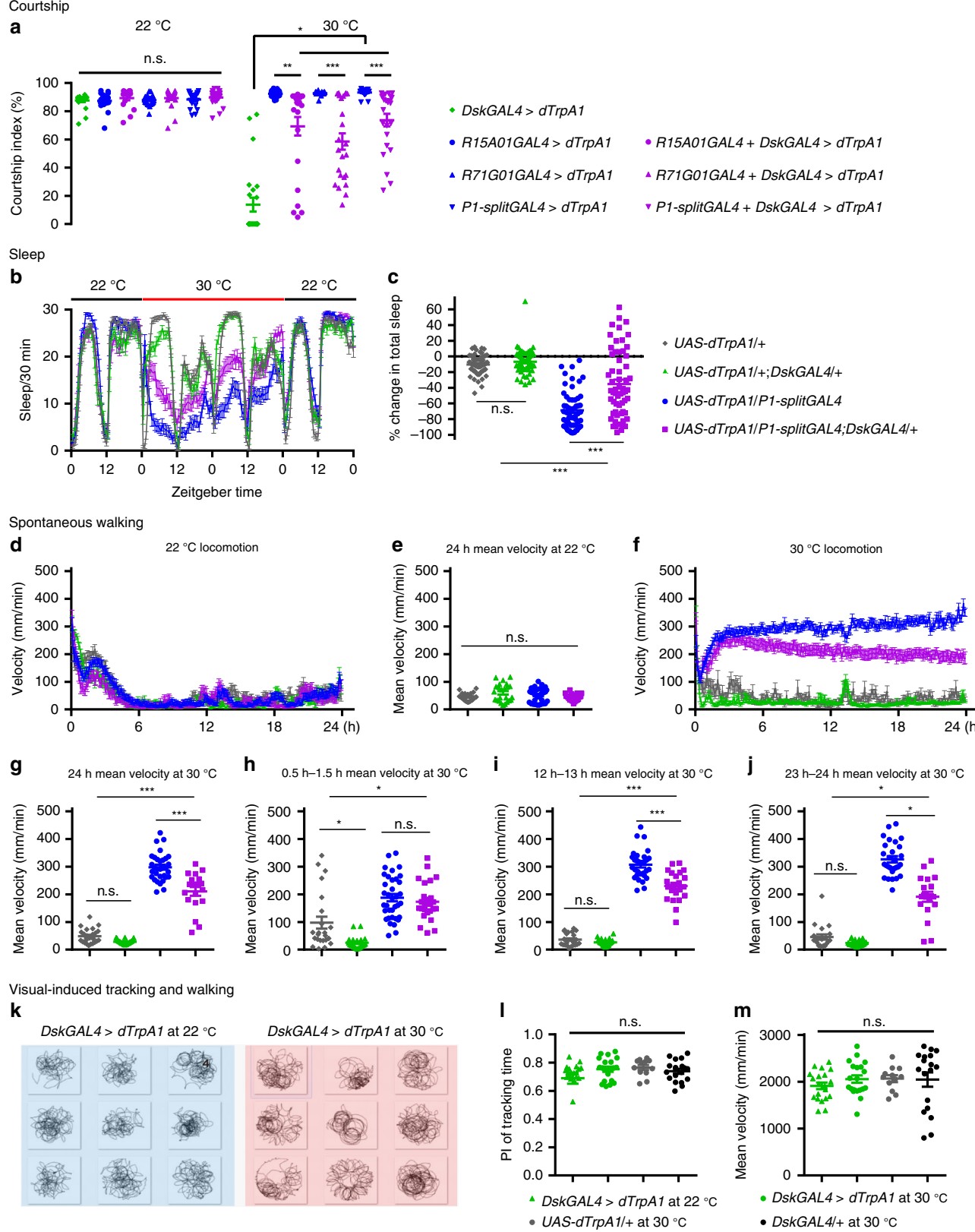

syb::spGFP1–10 in *DskGAL4* neurons and expression of *CD4::spGFP11* in *R15A01LexA* labeled P1 neurons, and observed no GFP signal (Fig. 5c, d). As *dsx^LexA* labels much more neurons involved in male courtship including P1 neurons, we repeated such activity-dependent GRASP experiment using *DskGAL4* and *dsx^LexA*, and did not observe any signal either (Fig. 5e). As a

positive control to validate this technique, we observed substantial GRASP signal using pan-neuronal driver *R57C10LexA* (Fig. 5f). These results indicate that there is no direct synaptic connection from *Dsk* neurons to P1 neurons, but do not exclude the possibility that secreted DSK peptides might act on P1 neurons as long as there are DSK receptors expressed there. In fact, P1 neurons do

**Fig. 4** *Dsk* and P1 neurons function antagonistically to regulate male sexual arousal. **a** Co-activation of *Dsk* and P1 neurons results in intermediate courtship levels. $n = 24, 19, 17, 24, 24, 20, 24, 24, 20, 24, 24, 24, 24,$ and 24 respectively (from left to right). Genotypes as indicated. $p < 0.05$, Kruskal–Wallis test. $*p < 0.05$, $**p < 0.01$, $***p < 0.001$, and $p > 0.1$ (n.s.) post hoc Dunn's multiple comparisons test. **b**, **c** *Dsk* and P1 neurons oppositely regulate sleep. $n = 60,$ 56, 57, and 60 respectively. $p < 0.001$, Kruskal–Wallis test. $***p < 0.001$ and $p > 0.99$ (n.s.), post hoc Dunn's multiple comparisons test. **d**, **e** Spontaneous walking velocity of individual males at permissive temperature (22 °C). $n = 24, 24, 36,$ and 19, respectively. $p = 0.07$, One-way ANOVA. **f–j** *Dsk* and P1 neurons oppositely regulate spontaneous walking velocity. $n = 24, 24, 32,$ and 19, respectively. For overall mean velocity (**g**): $p < 0.001$, One-way ANOVA. $***p < 0.001$ and $p > 0.1$ (n.s.), post hoc Tukey's multiple comparisons test. For mean velocity from 0.5 h to 1.5 h (**h**): $p < 0.001$, Kruskal–Wallis test. $*p < 0.05$ and $p > 0.99$ (n.s.), post hoc Dunn's multiple comparisons test. For mean velocity from 12 h to 13 h (**i**): $p < 0.001$, One-way ANOVA. $***p < 0.001$ and $p > 0.1$ (n.s.), post hoc Tukey's multiple comparisons test. For mean velocity from 23 h to 24 h (**j**): $p < 0.001$, Kruskal–Wallis test. $*p < 0.05$ and $p > 0.99$ (n.s.), post hoc Dunn's multiple comparisons test. **k** Representative walking trajectories in a visual-induced walking assay during a 6-min observational time. **l**, **m** Activation of *Dsk* neurons does not affect visual tracking ability or visual-induced walking speed. $n = 20, 20, 12,$ and 18, respectively. For visual tracking performance: $p = 0.3265$, Kruskal–Wallis test; for walking speed: $p = 0.3544$, Kruskal–Wallis test. n.s. not significant. Error bars indicate SEM. Source data are provided as a Source Data file

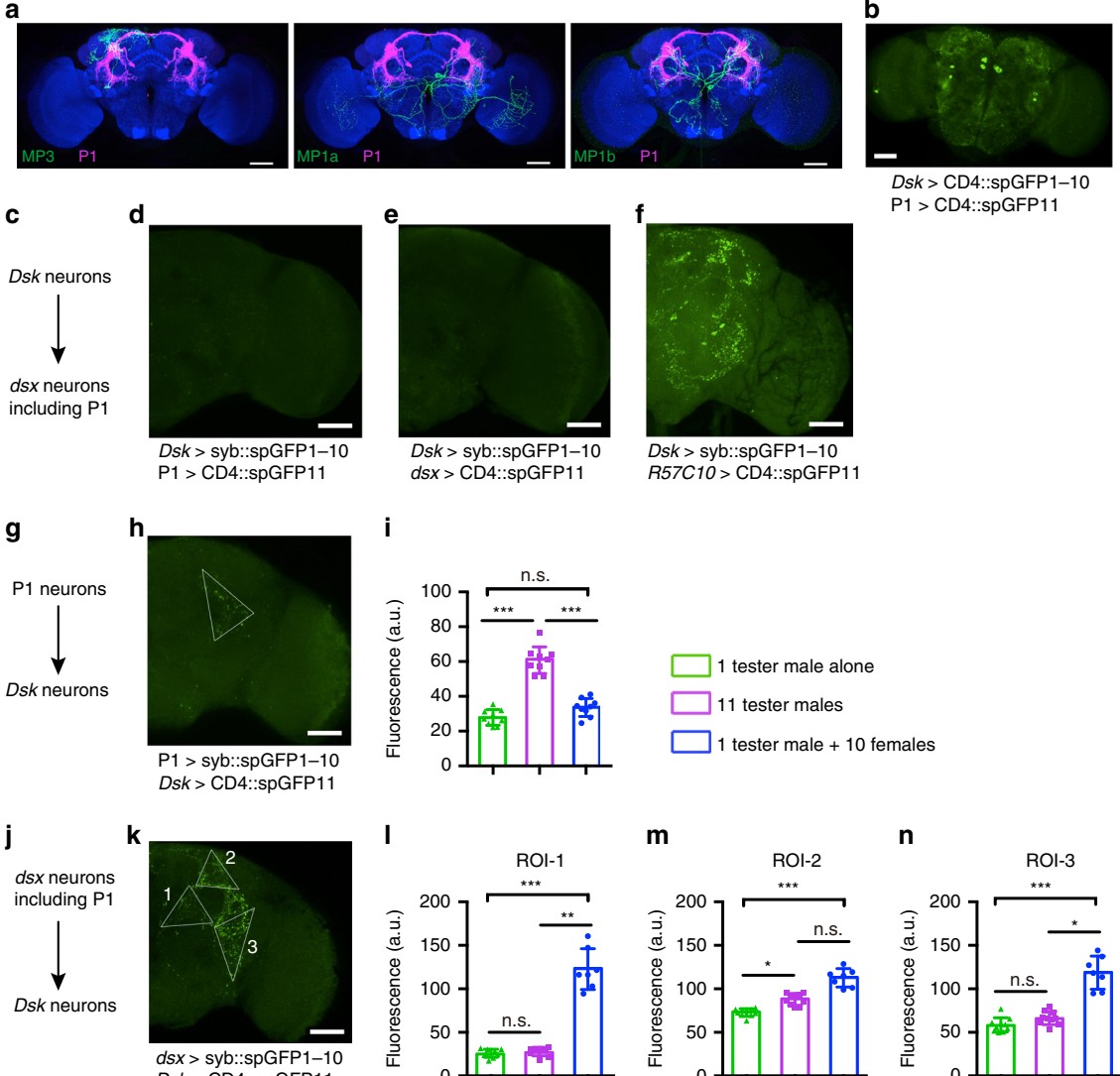

**Fig. 5** Experience-dependent synaptic interaction between *Dsk* and P1 neurons. **a** Registration of P1 neurons and *Dsk* MP neurons in a standard brain. **b** Potential membrane contacts between P1 and *Dsk* neurons as revealed by conventional GRASP technique. Representative of five male brains. **c–f** There is no syb-GRASP signal from *Dsk* neurons to *dsx* neurons including P1 neurons. Representative of five male brains for each genotype. **g–i** Experience-dependent synaptic transmission from P1 neurons to *Dsk* neurons as revealed by syb-GRASP signals. $n = 9$ for each. $p < 0.001$, One-way ANOVA. $***p < 0.001$, post hoc Tukey's multiple comparisons test. **j–n** Experience-dependent synaptic transmission from *dsx* neurons (including P1 neurons) to *Dsk* neurons as revealed by syb-GRASP signals in single-housed males (green bar) and group housed males (magenta bar for male–male group, and blue bar for male–female group). $n = 10, 10$ and 7 for each group respectively. For ROI-1: $p < 0.001$, Kruskal–Wallis test. $**p < 0.01$, $***p < 0.001$ and $p > 0.99$ (n.s.), post hoc Dunn's multiple comparisons test. For ROI-2: $p < 0.001$, Kruskal–Wallis test. $*p < 0.05$, $***p < 0.001$ and $p = 0.08$ (n.s.), post hoc Dunn's multiple comparisons test. For ROI-3: $p < 0.001$, Kruskal–Wallis test. $*p < 0.05$, $***p < 0.001$ and $p = 0.31$ (n.s.), post hoc Dunn's multiple comparisons test. n.s. not significant. Error bars indicate SEM. Source data are provided as a Source Data file

express DSK receptor, and could be regulated by DSK, but see below for details.

We then tested if P1 neurons might be presynaptic to *Dsk* neurons by driving expression of *syb::spGFP1–10* in *R15A01LexA* labeled P1 neurons and expression of *CD4::spGFP11* in *DskGAL4* neurons, and observed reconstituted GFP signal specifically in the lateral protocerebral complex region (Fig. 5g, h), indicating that P1 neurons form direct synaptic connections with *Dsk* neurons. As such reconstituted GRASP signals are activity dependent, we compared these GRASP signals in males that have different housing experiences. We found that synaptic transmission from P1 to *Dsk* neurons, represented by the intensity of GRASP signal, are much stronger in male–male group-housed conditions (11 males), compared to single-housed and male–female group-housed (1 male + 10 females) (Fig. 5i). These results not only providing evidence of direct synaptic transmission from P1 neurons to *Dsk* neurons, but also indicating that such synaptic transmission is experience dependent and more efficiently induced by male–male group housing. We then tested the GRASP signal from all *dsx* neurons (including P1) to *Dsk* neurons (*syb::spGFP1–10* driven by *dsx^LexA* and *CD4::spGFP11* driven by *DskGAL4*), and observed much more GRASP signals where they were divided into three parts as ROI-1, ROI-2, and ROI-3 for statistics (Fig. 5j, k). Comparing the intensity of GRASP signals under the above three housing conditions, we found, in all ROIs, GRASP signals were significantly enhanced by housing a male with ten females, where only in ROI-2 (overlaps with GRASP signals that from P1 to *Dsk*, see Fig. 5h) GRASP signals were significantly enhanced by male–male group housing, suggesting distinct properties of synaptic transmission from different *dsx* (e.g., P1 vs. pCd and pC2) neurons to *Dsk* neurons. We tried *LexA* drivers labeling pCd (*R41A01-LexA*) or pC2 (*R40F04-LexA*), as well as many other neurons (we do not have clean drivers for pCd and pC2 yet), but did not observe any GRASP signal from these *LexA* labeled neurons to *Dsk* neurons, probably due to the weak labeling of pCd and pC2 neurons by these *LexA* lines. Generating better genetic tools to label subsets of *dsx* neurons will help to understand how different populations of *dsx* neurons interact with *Dsk* neurons to regulate male courtship. Together these results indicate that *Dsk* neurons receive direct synaptic transmission, in an experience-dependent manner, from *dsx* neurons including P1 neurons.

**DSK peptides inhibits male courtship behavior**. The above results establish that four pairs of *Dsk*- and *fru^M*-expressing neurons function antagonistically with P1 neurons to regulate male courtship, but whether these *Dsk* neurons function through DSK peptides is unclear. The *Dsk* gene encodes three mature peptides DSK-0, DSK-1 and DSK-2 (Fig. 6a), two of which (DSK-1 and DSK-2) are Cholecystokinin (CCK)-like peptides[39,47]. We generated two deletion alleles of *Dsk* using the CRISPR/Cas9 technique[48,49], which we termed $\Delta Dsk^1$ and $\Delta Dsk^2$ (Fig. 6a). $\Delta Dsk^1$ leads to deletion of all three peptides, and $\Delta Dsk^2$ only results in mutation of the DSK-2 peptide. Indeed, we observed no anti-DSK signal in homozygous $\Delta Dsk^1$ flies, and weaker signals in $\Delta Dsk^1/\Delta Dsk^2$ flies, as the DSK antibody recognizes both DSK-1 and DSK-2 (Fig. 6b). These *Dsk* mutant males are fully viable and fertile, and court vigorously to females like control males (CIs > 80%, Fig. 6c). We also checked male–male courtship and did not observe obvious phenotype (Fig. 6d). These results indicate that loss of DSK in an otherwise wild-type fly does not obviously affect male courtship behaviors.

To determine whether release of DSK peptides from *Dsk* neurons is responsible for courtship inhibition, we activated *Dsk* neurons in the above *Dsk* mutant background. We used

homozygous $\Delta Dsk^1$ males as the null mutant for all three DSK peptides, and its combination with $\Delta Dsk^2$ as well as homozygous $\Delta Dsk^2$ as specific mutants for the DSK-2 peptide. We found that courtship inhibition by activation of *Dsk* neurons is fully dependent on secretion of DSK peptides, particularly the DSK-2 peptide, as loss of DSK-2 fully restores courtship of *DskGAL4/UAS-dTrpA1* males at 30 °C (CI > 80%), comparable to wild-type courtship (Fig. 6e). These results indicate that DSK-2 is indispensable for courtship inhibition in males with *Dsk* neurons being activated, and further evidences are needed to reveal the role of DSK-1 in courtship inhibition. Furthermore, we knocked-down *Dsk* using RNA interference (RNAi)[50,51], which significantly decreased DSK immunoreactivity (Fig. 6f, g), and found that courtship behavior was also restored in *DskGAL4/UAS-dTrpA1* males at 30 °C (CI > 80%, Fig. 6h). We also found that activation of *Dsk* neurons in *DskGAL4/UAS-TrpA1* males at 30 °C for 30 min decreased DSK immunoreactivity in many parts of *Dsk* neurons including soma, suggestive of DSK secretion in response to neuronal activation (Supplementary Fig. 6). Taken together these results demonstrated that DSK secretion from *Dsk* neurons is responsible for courtship inhibition in males with activated *Dsk* neurons.

As we did not observe increased courtship in *Dsk* mutant males, probably due to already high levels of male–female courtship or potential compensation to the loss of DSK modulation, we set out to test whether acutely increased expression of DSK in an otherwise wild-type fly would inhibit male courtship. We utilized a newly invented technique using Cas9 transcriptional activators to activate gene expression[52,53], and found that acutely over expression of *Dsk* two days before courtship assay, using temperature dependent *tub-GAL80^ts*, significantly decreased male courtship, compared to both genetic and temperature controls (Fig. 6i). Together these results indicate that DSK peptides released from *Dsk* neurons inhibit male courtship behavior.

**DSK inhibits male courtship through its receptor CCKLR-17D3**. DSK peptides have two G-protein coupled receptors: the CCK-like receptor (CCKLR) at 17D1 and 17D3[54,55]. We found that Loss of *CCKLR-17D1* using a deletion mutant *Df(1)Exel9051* (*Δ17D1*) did not restore courtship levels when activating *Dsk* neurons (Fig. 7d). We then generated a deletion mutation of *CCKLR-17D3* (*Δ17D3*) by CRISPR/Cas9 tools[48,49]. This deletion removes the start codon and the coding sequence of *CCKLR-17D3* (Fig. 7a–c). We found that the courtship-inhibiting effect of activating *Dsk* neurons was abolished in *Δ17D3* males (Fig. 7d), indicating that DSK functions through its receptor CCKLR-17D3 but not CCKLR-17D1 to inhibit male courtship.

To further confirm the role of *CCKLR-17D3* on male courtship inhibition, we used the same technique[52,53] as above activating *Dsk* expression to activate *CCKLR-17D3* expression, and found that acutely over expression of *CCKLR-17D3* two days before courtship assay significantly decreased male courtship behavior (Fig. 7e). In contrast, over expression of *CCKLR-17D1* did not significantly affect male courtship (Fig. 7e). These results provide further evidence that DSK/CCKLR-17D3 signaling inhibits male courtship behavior.

**DSK/CCKLR-17D3 signaling responds to past experiences**. In order to further understand the physiological role that DSK signaling plays, we measured the level of DSK and CCKLR-17D3 expression using quantitative real-time PCR, and found that DSK/CCKLR signaling responds to multiple physiological conditions, including re-feeding, aging, group-housing and P1 neuronal activation (Supplementary Fig. 7). Since we also found that

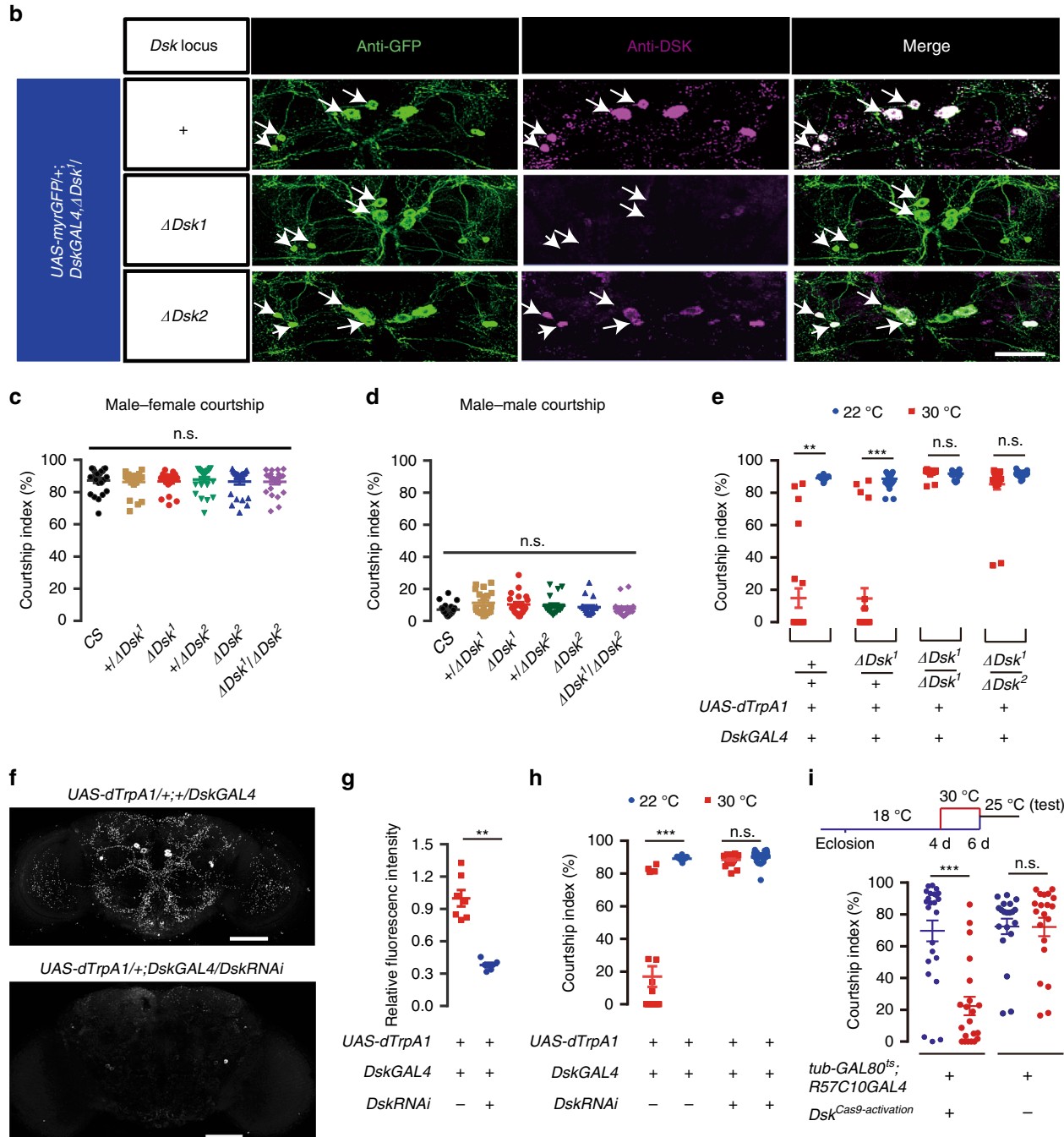

**a**

>Dsk (wild-type)

MGPRSCTHFATLFMPLWALAFCFLVVLPIPAQTTSLQNAKDDRRLQELESKIGGEIDQPIANLVGPSFSLFGDRR<u>NQKTMSF</u>

DSK-0

GRRVPLISRPIIPIELDLLMDNDDERTKAKR<u>FDDYGHMRF</u>GKR<u>GGDDQFDDYGHMRF</u>GR.

DSK-1          DSK-2

>ΔDsk¹

MGPRSCTHFATLFMPLWALAFCFLVVLPIPAQTTSLQNAKDDRRLQELESKIGGEIDQPIANLVGPSGVIRKQ.

>ΔDsk²

MGPRSCTHFATLFMPLWALAFCFLVVLPIPAQTTSLQNAKDDRRLQELESKIGGEIDQPIANLVGPSFSLFGDRRNQKTMSF

GRRVPLISRPIIPIELDLLMDNDDERTKAKRFDDYGHMRFGKRGGDDQFDDYTLAISLVI.

**b**

*UAS-myrGFP/+; DskGAL4,ΔDsk/*

| *Dsk* locus | Anti-GFP | Anti-DSK | Merge |
|---|---|---|---|
| + | | | |
| ΔDsk1 | | | |
| ΔDsk2 | | | |

**c** Male–female courtship — n.s.

Courtship index (%): CS, +/ΔDsk¹, ΔDsk¹, +/ΔDsk², ΔDsk², ΔDsk¹/ΔDsk²

**d** Male–male courtship — n.s.

Courtship index (%): CS, +/ΔDsk¹, ΔDsk¹, +/ΔDsk², ΔDsk², ΔDsk¹/ΔDsk²

**e** • 22 °C ■ 30 °C

** *** n.s. n.s.

| | $\frac{+}{+}$ | $\frac{\Delta Dsk^1}{+}$ | $\frac{\Delta Dsk^1}{\Delta Dsk^1}$ | $\frac{\Delta Dsk^1}{\Delta Dsk^2}$ |
|---|---|---|---|---|
| UAS-dTrpA1 | + | + | + | + |
| DskGAL4 | + | + | + | + |

**f**

*UAS-dTrpA1/+;+/DskGAL4*

*UAS-dTrpA1/+;DskGAL4/DskRNAi*

**g**

Relative fluorescent intensity — **

| UAS-dTrpA1 | + | + |
|---|---|---|
| DskGAL4 | + | + |
| DskRNAi | − | + |

**h** • 22 °C ■ 30 °C

*** n.s.

| UAS-dTrpA1 | + | + | + | + |
|---|---|---|---|---|
| DskGAL4 | + | + | + | + |
| DskRNAi | − | − | + | + |

**i**

18 °C → 30 °C → 25 °C (test)
Eclosion    4 d  6 d

*** n.s.

| tub-GAL80ᵗˢ; R57C10GAL4 | + | + |
|---|---|---|
| Dsk^Cas9-activation | + | − |

group-housing enhanced synaptic transmission from P1 neurons to *Dsk* neurons according to the activity-dependent GRASP signals (see Fig. 5i), we asked whether group-housing may affect male courtship in a DSK-dependent manner. We first tested male–female courtship by single-housed males or group-housed

males (11 males, the same as used in Fig. 5i), and found no difference of male courtship in wild-type control males and *Dsk* knocked-down males (*R57C10GAL4/UAS-DskRNAi*) (Supplementary Fig. 8a). We then repeated male courtship, but tested in restricted conditions (headless females as targets under dark

**Fig. 6** DSK peptides inhibit male courtship behavior. **a** Predicted amino acid sequence and mature peptides from the *Dsk* gene in wild-type, *ΔDsk*1 and *ΔDsk*2 flies. **b** Validation of *ΔDsk* mutants by anti-DSK staining. Representative of 5 male brains each. Arrows indicate cell bodies of MP neurons. Scale bars, 50 μm. **c** *Dsk* mutant males have normal courtship toward virgin females. Genotypes as indicated. $n = 24$ for each. $p = 0.22$, Kruskal–Wallis test. **d** *Dsk* mutant males do not show abnormal male–male courtship. $n = 20$ for each. $p = 0.42$, Kruskal–Wallis test. **e** Courtship inhibition by activating *DskGAL4* neurons is dependent on the DSK-2 peptide. $n = 24$ for each. $p < 0.001$, Kruskal–Wallis test. **p < 0.001, ***p < 0.001 and $p > 0.1$ (n.s.), post hoc Dunn's multiple comparisons test. **f** Down-regulation of *Dsk* gene expression using *UAS-DskRNAi* as revealed by anti-DSK staining in males raised at 22 °C. Scale bars, 50 μm. **g** Quantification of anti-DSK signal in the brain. $n = 7$ and 6. ***p < 0.01, Mann–Whitney U test. **h** Courtship inhibition by activating *DskGAL4* neurons is dependent on DSK, as knocking down *Dsk* by RNAi restores courtship by *DskGAL4/UAS-dTrpA1* males at 30 °C. $n = 24$ for each. $p < 0.001$, Kruskal–Wallis test. ***p < 0.001 and $p = 0.85$ (n.s.), post hoc Dunn's multiple comparisons test. **i** Acutely over expression of DSK peptides two days before courtship test significantly suppresses male courtship. $n = 24, 21, 20,$ and 20, respectively. $p < 0.001$, Kruskal–Wallis test. ***p < 0.001 and $p > 0.99$ (n.s.), post hoc Dunn's multiple comparisons test. n.s. not significant. Error bars indicate SEM. Source data are provided as a Source Data file

condition) to reduce redundancy of sensory stimulus on courtship[56], and found that group-housing significantly reduced male courtship under these conditions; however, such group-housing induced courtship reduction is not observed in *Dsk* knocked-down males, as they courted equally high under two rearing conditions (Supplementary Fig. 8b). These results indicate that DSK signaling responds to multiple physiological states and past experiences, and at least male–male group-housing experience increases DSK expression and reduces male courtship.

**CCKLR-17D3 in *fru^M* neurons inhibits male courtship**. We showed that DSK functions in four pairs of *fru^M*-positive neurons to inhibit male courtship, and acts through its receptor CCKLR-17D3, but where CCKLR-17D3 functions to inhibit male courtship in unknown. As we failed to generate a functional CCKLR-17D3 antibody, we made a knock-in *GAL4* into the *CCKLR-17D3* locus (*17D3^GAL4*) to recapitulate its expression (Fig. 7f). We found that *17D3^GAL4* drives expression broadly in the brain and the ventral nerve cord (Fig. 7g). To test whether *17D3^GAL4* drives expression in *fru^M* neurons, we used an intersectional strategy to label overlapping neurons between *17D3^GAL4* and *fru^LexA*. Interestingly, the intersectional neurons include the male-specific P1 neurons that are crucial for courtship initiation, and the mushroom bodies that are important for sleep and locomotion, as well as many other neurons (Fig. 7h, i). That *17D3^GAL4* drives expression in P1 neurons is further confirmed by two other intersectional labeling experiments (Fig. 7j, k). Thus we have found a neuropeptide signaling that functions in the sex circuitry: *Dsk* MP neurons receive direct synaptic inputs from many *dsx* neurons including P1 neurons, and then act on many *fru^M* and/or *dsx* neurons (including P1) via secretion of DSK peptides to fulfill their function (Fig. 7l). To further test the role of CCKLR-17D3, we knocked down its expression using RNAi in all *fru^M*-expressing (*fru^GAL4*), all *dsx*-expressing (*dsx^GAL4*), or a subset of P1 neurons (*P1-splitGAL4*, ~10 pairs of P1 neurons), but found no difference for male courtship (Supplementary Fig. 9a). We then activated CCKLR-17D3 expression using above mentioned tools[53,57] specifically in *fru^GAL4*, *dsx^GAL4*, or *P1-splitGAL4* labeled neurons, and found that over expression of CCKLR-17D3 in all *fru^M* or *dsx* neurons, but not P1 neurons alone, significantly decreased male courtship (Supplementary Fig. 9b). These results indicate that CCKLR-17D3 functions in many *fru^M* and/or *dsx* neurons (at least more than ~10 pairs of P1 neurons) to inhibit male courtship.

**DSK signaling inhibits both male and female sexual behaviors**. As we showed above that activating *Dsk* neurons also inhibited virgin female receptivity, we asked whether such inhibition in female sexual behavior depends on DSK/CCKLR-17D3 signaling. We found that suppression of female receptivity by activating *DskGAL4* neurons also depends on DSK-2 and CCKLR-17D3 as

loss of DSK-2 (*UAS-dTrpA1/+; DskGAL4 ΔDsk1/ΔDsk2*, Supplementary Fig. 10a) or CCKLR-17D3 (*Δ17D3; UAS-dTrpA1/+; DskGAL4/+*, Supplementary Fig. 10b) restores female receptivity to levels comparable to control females. Note that there is no P1 neuron in the female brain such that the P1-*Dsk* neuronal interaction is male-specific and does not account for female behaviors. As *dsx* neurons, particularly pC1 and pCd *dsx* neurons are important for virgin female receptivity[58], we tested whether *dsx* neurons have direct synaptic transmission to *Dsk* neurons, as found in males (Fig. 5j–n). When *syb::spGFP1–10* is driven by *dsx^LexA* and *CD4::spGFP11* driven by *DskGAL4*, we observed substantial GRASP signals in a ring-like shape in the lateral protocerebral complex that are important for sexual behaviors, and some other regions in female brains (Supplementary Fig. 10c). These results clearly show that *Dsk* neurons interact with sexually dimorphic *dsx* neurons in both sexes to suppress sexual behaviors.

**Discussion**

Our results identify, at single-neuron resolution, four pairs of *fru^M*-expressing *Dsk* neurons (MP1 and MP3) that suppress male and female sexual behaviors. The suppression of male and female sexual behaviors depends on the secretion of the neuropeptide DSK-2, which then acts on one of its receptors CCKLR-17D3 that is expressed in many *fru^M* neurons including P1 neurons and the mushroom bodies. *Dsk* neurons function antagonistically with courtship promoting P1 neurons to co-regulate male courtship, as well as sleep and spontaneous walking.

Cholecystokinin (CCK) signaling appears well conserved over evolution and modulate multiple behaviors[47]. In *Drosophila*, the CCK-like sulfakinin (DSK) is multifunctional and has been reported to be involved in regulating aspects of food ingestion and satiety[40], aggression[59], as well as escape-related locomotion and synaptic plasticity during neuromuscular junction development[55]. In mammals, CCK generated from the intestine acts on its receptors in the nucleus of the solitary tract of the brain to transmit satiety signaling and thus inhibit feeding[60]. Furthermore, CCK signaling in the nucleus accumbens modulates dopaminergic influences on male sexual behaviors in rats[61]. CCK is also involved in nociception, learning and memory, aggression and depressive-like behaviors[62–64].

Despite its significant and conserved roles in modulating multiple innate and learned behaviors, how CCK or DSK signaling responds to environment and/or internal changes, and acts on specific neurons expressing its receptors to modulate multiple behaviors, is still rarely known. Our finding that DSK/CCKLR signaling functions in the *fru^M*- and/or *dsx*-expressing sex circuitry to inhibit male courtship is an effort to use *Drosophila* as a model to investigate how this conserved signaling modulates animal behaviors.

Our results uncovered a functional circuitry from many *dsx* neurons (including courtship-promoting P1 neurons) to four

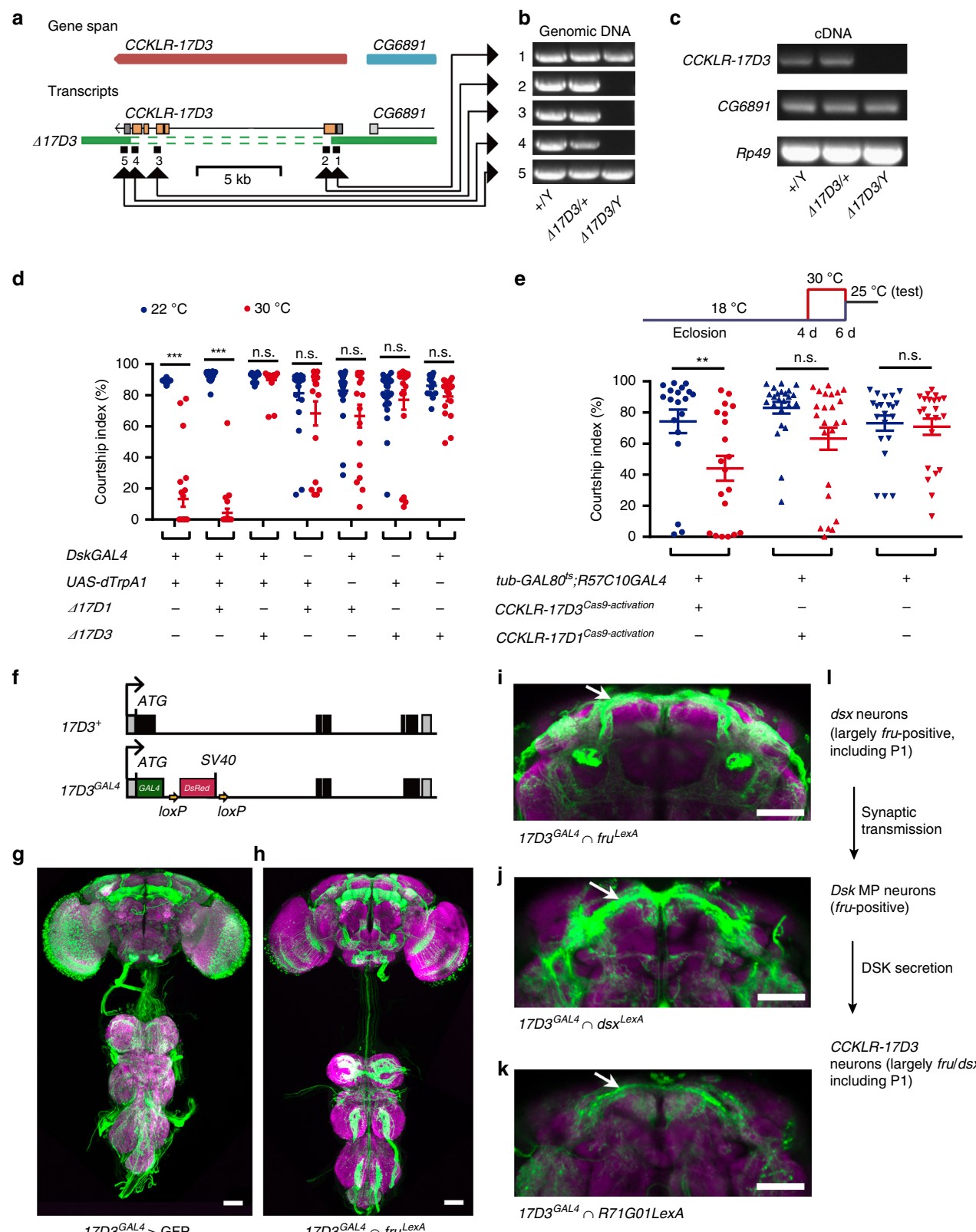

pairs of *Dsk* MP neurons via direct synaptic transmission, and these MP neurons then modulate *CCKLR-17D3* neurons including many *fruM* and/or *dsx* neurons via secretion of DSK peptides. It is of particular interest to reveal how the four pairs of MP neurons integrate sensory information (in any), physiological

states and past experiences in the future to better understand how this neuropeptide signaling modulate arousal states. We still do not know if these MP neurons receive sensory inputs, but since they receive inputs from many *dsx* neurons including P1 neurons that integrate multiple chemosensory information[24,25], these

**Fig. 7** CCKLR-17D3 inhibits male courtship. **a–c** Generation and validation of a 9.04 kb deletion mutant of the *CCKLR-17D3* gene. **d** Courtship inhibition by activating *DskGAL4* neurons is dependent on DSK's receptor CCKLR-17D3 but not CCKLR-17D1, as mutation in *CCKLR-17D3* but not in *CCKLR-17D1* restores courtship by *DskGAL4/UAS-dTrpA1* males at 30 °C. Genotypes as indicated. $n = 24, 24, 24, 24, 24, 24, 18, 24, 18, 24, 24, 24, 18,$ and 13, respectively. ***$p < 0.001$, Mann–Whitney U test. **e** Acutely over expression of CCKLR-17D3 but not CCKLR-17D1 two days before courtship test significantly suppresses male courtship. $n = 19, 20, 24, 24, 21,$ and 22 respectively. $p < 0.01$, Kruskal–Wallis test. **$p < 0.01$ and $p > 0.1$ (n.s.), post hoc Dunn's multiple comparisons test. n.s. not significant. Error bars indicate SEM. **f** Generation of knock-in *GAL4* into the *CCKLR-17D3* locus. **g** Expression pattern of *17D3^GAL4* visualized by *UAS-myrGFP* in male CNS. Representative of eight males. **h, i** Intersectional expression between *17D3^GAL4* and *fru^LexA* in the male CNS. Representative of five males. Arrow indicates the male-specific P1 neurons (**i**). Genotype: *17D3^GAL4/Y; UAS > stop > myrGFP LexAop2-FlpL/+; fru^LexA/+*. **j, k** Intersectional expression between *17D3^GAL4* and *dsx^LexA* (**j**), *17D3^GAL4* and *R71G01LexA* (**k**) in males. Representative of five males each. Arrow indicates the male-specific P1 neurons. Scale bars, 50 μm. **l** An illustration from *dsx* neurons acting on *Dsk* neurons via synaptic transmission, and *Dsk* neurons to *CCKLR-17D3* neurons via secretion of DSK. Source data are provided as a Source Data file

sensory inputs will at least relay to *Dsk* MP neurons via P1. Whether there are other pathways from sensory inputs to MP neurons awaits further study. We also note that multiple physiological changes including feeding states, aging and sleep deprivation, as well as past housing conditions affect expression of DSK/CCKLR-17D3, but how they affect DSK signaling and behaviors is still unclear. We showed that male–male group-housing increases DSK expression and thereby reduces male courtship at least under a restricted condition, and previous findings also revealed opposite effects of group-housing on the excitability of P1 neurons in males that have *fru^M* function or lack *fru^M* function[65,66]. Such male–male housing experience may mildly reduce sexual arousal in a persistent manner, perhaps by increasing DSK expression, but how such housing condition affects physiological roles of *Dsk* MP neurons and P1 neurons awaits further functional imaging studies on a potential P1-*Dsk*-P1 functional loop (and a much complex *dsx*-*Dsk*-17D3 pathway) with sensory stimulation under different physiological states. In terms of the time scale that DSK functions to inhibit male courtship, our results indicate an immediate behavioral effect upon *Dsk* neuronal activation. We also note that activation of *Dsk* neurons inhibits male courtship and lasts for minutes, and previous findings also showed that activation of P1 neurons promoted wing extension and aggression and lasts for minutes[42,65]. These persistent behavioral effects may represent persistent arousal states regulated by *Dsk* and P1 neurons in this study; however, how this persistency is generated both in the circuit level and behavioral level still needs further investigation.

*Dsk* mutants do not have obvious courtship abnormality under our courtship assays. There are at least two possibilities: (1) DSK may function only in specific conditions (e.g., group-housing, as shown in Supplementary Fig. 8) that increase its expression to inhibit courtship; and (2) There are redundant inhibitory signals for courtship, such as another neuropeptide SIFamide that acts on *fru^M* neurons[30,31], although they specifically inhibit male–male courtship. Further studies on how DSK/CCKLR signaling is activated under certain conditions, as well as how DSK, SIFamide and other inhibitory signals (if any) jointly modulate male courtship are needed to fully understand this. Nevertheless, that courtship inhibition by activation of *Dsk* neurons depends on DSK/CCKLR-17D3 signaling, and increasing such signaling through Cas9 activators in an otherwise wild-type male efficiently inhibited courtship, unambiguously reveal the role of DSK/CCKLR-17D3 signaling in suppressing sexual behaviors.

As DSK signaling modulates multiple behaviors, one may argue that its role in male courtship is not specific, e.g., activation of *Dsk* neurons may drive a competing behavior that phenotypically shunting male courtship. Although we cannot exclude such possibility, we listed a number of evidences as also summarized in Fig. 8 to support DSK's role with specificity in courtship inhibition: (1) DSK functions in four pairs of *fru^M*-expressing neurons to inhibit courtship; (2) males with activated

*Dsk* neurons rarely court virgin females, while they follow rotating visual objects normally; (3) *Dsk* neurons receive synaptic transmission from courtship promoting P1 neurons (and many other *dsx*-expressing neurons) in an experience-dependent manner; (4) *Dsk* and P1 neurons antagonistically modulate sexual behaviors and wakefulness; and (5) DSK receptor CCKLR-17D3 inhibits male courtship and expresses in many *fru^M* and/or *dsx* neurons including P1 neurons. We note that CCKLR-17D3 is expressed broadly in the CNS including not only P1 neurons, but also mushroom bodies that regulate a range of behaviors including learning, locomotion and sleep[67]. That DSK signaling is multifunctional is possibly due to broad expression of its receptors, and further studies on dissection of CCKLR function in subsets of neurons will help to understand how DSK/CCKLR signaling modulates multiple behaviors.

The decision for male flies to court or not depends on not only environmental cues such as availability and suitability of potential mates (males, virgin females, or mated females), but also their internal states (e.g., thirsty or sleepy). We propose that there are at least four factors affecting such a decision: (1) external cues that inhibit courtship, referred to as Ex-In factor, such as the male-specific pheromone cVA[4]; (2) external cues that are excitatory for courtship, referred to as Ex-Ex factor, such as courtship song[68]; (3) internal states that inhibit courtship, referred to as In-In factor; and (4) internal states that are excitatory for courtship, referred to as In-Ex factor. These factors dynamically change and jointly determine males' decision to court or not.

Substantial progress has been made on how Ex-In and Ex-Ex factors jointly modulate the activity of male-specific P1 neurons, which is crucial for courtship initiation[23–25]. In contrast, much less is understood on In-In and In-Ex factors. Recently, Zhang and colleagues found that dopaminergic modulation of P1 neurons drives male courtship not only by desensitizing P1 to inhibition, but also by promoting recurrent P1 stimulation[69,70], thus may act as an In-Ex factor for male courtship. Note that all the three factors mentioned above converge on P1 neurons, making them a decision-making center for male courtship. The DSK/CCKLR signaling we identified in this study is of particular interest, as it is likely to act as an In-In factor for male courtship, and above all, it does not simply act on P1 neurons like three other factors, but instead forms a potential functional loop with P1 neurons and antagonizes P1 function in modulating male courtship and wakefulness. That *Dsk* neurons receive synaptic transmission from P1 neurons and other *dsx*-expressing neurons in an experience-dependent manner further highlights a central role that the DSK/CCKLR signaling plays. These factors, excitatory vs. inhibitory, external vs. internal, jointly control appropriate performance of sexual behaviors, and further studies will reveal how P1 and other *dsx*-expressing neurons physiologically interact with *Dsk* neurons to balance behavioral output.

A prominent feature of the neuronal control of male and female sexual behaviors in *Drosophila* is that, despite large

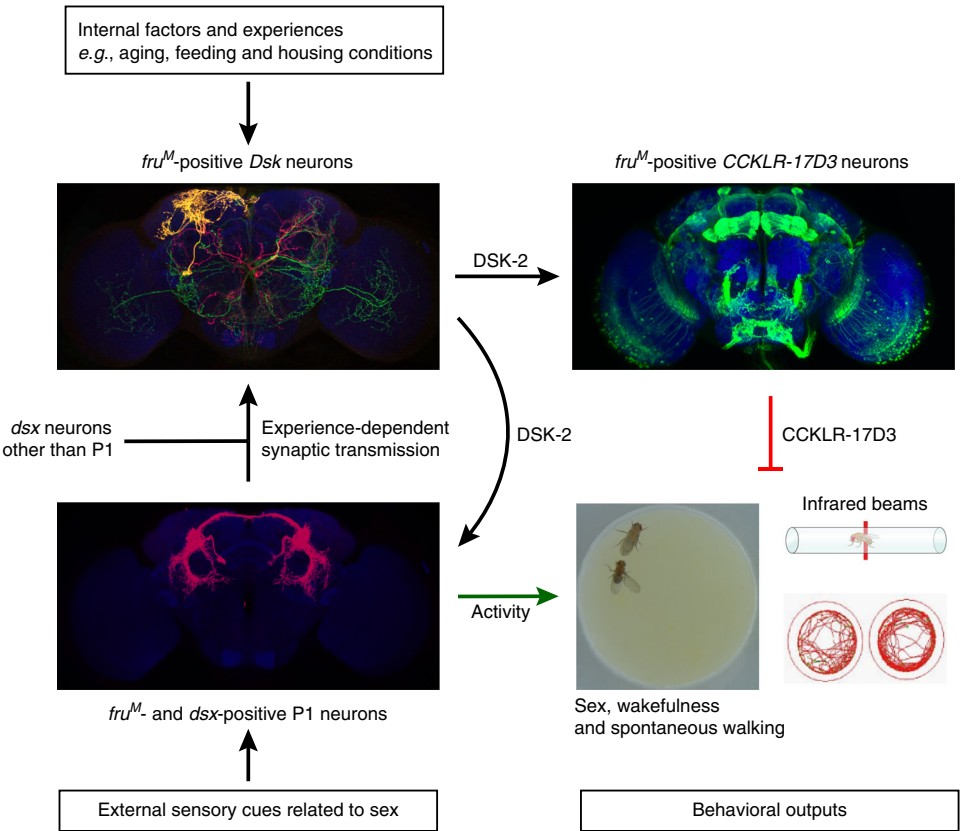

**Fig. 8** A model of DSK signaling regulating male sexual arousal. On one hand, male-specific P1 neurons that express both FRU$^M$ and DSX$^M$ integrate multiple sensory cues including both excitatory (e.g. from virgin females) and inhibitory (e.g. from males) information for courtship, and activation of P1 neurons persistently promotes male sexual behavior and wakefulness (reduction of sleep and increase of spontaneous walking), indicative of increased sexual arousal. On the other hand, four pairs of Dsk- and fru$^M$-expressing neurons interact with P1 neurons to form a potential functional loop, and act through DSK/CCKLR-17D3 signaling in fru$^M$ circuitry to persistently inhibit male sexual behavior and wakefulness. The DSK/CCKLR-17D3 signaling responds to multiple physiological conditions including aging, feeding and group-housing experiences. Thus the interplay between internal states represented by DSK/CCKLR-17D3 signaling and external sensory stimulation represented by P1 neuronal activity orchestrates appropriate male sexual behavior

similarity in sensory systems, central integrative neurons are sex-specific in the two sexes, with *dsx*-expressing pC1 neurons integrating olfactory and auditory cues and promoting receptivity to courting males[58], and *fru$^M$*-expressing P1 neurons (largely overlapped with *dsx*-expressing pC1) integrating olfactory, gustatory, and auditory cues and promoting courtship to females[24,25,68]. In contrast, the four pairs of *Dsk*-expressing MP neurons we investigated in this study are sexually monomorphic and inhibit both male courtship and female receptivity. Thus DSK/CCKLR signaling may inhibit sexual behaviors in response to physiological changes that are common to both sexes, while sex-promoting central neurons integrating distinct sensory cues are sexually dimorphic. Interestingly, these *Dsk* neurons common to both sexes receive synaptic transmission from sexually dimorphic *dsx* neurons in both males and females, providing a simple solution to link sex-specific excitatory and sexually non-specific inhibitory control of sexual behaviors in males and females.

## Methods

**Fly stocks**. Flies were maintained at 22 °C or 25 °C in a 12 h:12 h light:dark cycle. Canton-S flies were used as the wild-type strain. *Neuropeptide-GAL4* lines[38] used in Fig. 1b, *UAS-dTrpA1*[36], *Δ17D1 [Df(1)Exel9051]*[55], *UAS-CD4::spGFP*[1-10] *LexOp-CD4::spGFP*1145, *UAS-CD4::spGFP*[11] *LexOp-syb::spGFP*[1-10], and *LexOp-CD4::spGFP*[11] *UAS-syb::spGFP*1–10 [46] have been described previously and are from Bloomington Stock Center. Two *SIFaGAL4* lines[71] were kindly provided by Dr. Yi Rao. *pBDPGAL4u*, *UAS-CsChrimson-mVenus*[41], *UAS > stop > CsChrimson-*

*mVenus*[68], *UAS-mCD8RFP*, *UAS-myrGFP*, *UAS-Stinger-GFP*, *LexOp-RedStinger*, *LexOp2-myrGFP*, *LexOp2-FlpL*, *UAS > stop > myrGFP*, *fru$^{GAL4}$*15 and *fru$^{LexA}$*72 have been described previously[73,74] and are obtained from Janelia Research Campus. *UAS-DskRNAi* was a gift from Tsinghua Fly Center (THU2073) at the Tsinghua University[50,51]. *DskGAL4* (attP2), *DskLexA* (attP2), *DskFlp* (attP2), *ΔDsk*1, *ΔDsk*2, *Δ17D3*, *17D3$^{GAL4}$*, *Dsk$^{Cas9-activation}$*, *CCKLR-17D3$^{Cas9-activation}$*, and *CCKLR-17D1$^{Cas9-activation}$* are generated in this study and described below.

**Genomic DNA and cDNA amplifications**. Genomic DNA was extracted from whole fly body in extraction buffer (0.1 M Tris-HCl, pH 9.0; 0.1 M EDTA; 1% SDS). Briefly, 30 flies were thoroughly crushed in extraction buffer with pestle. After incubation at 70 °C for 30 min, samples were lysed by adding potassium acetate to the final concentration of 800 mM, re-incubated on ice for 30 min. Precipitation was pelleted by centrifugation at 12,000$g$ for 15 min, and the supernatant was collected. Half the volume of isopropanol was added to the supernatant, and precipitation was pelleted by centrifugation for 5 min. The pellet was washed with ice-cold 70% ethanol once, and the pellet was dried before resuspended in 100 μl of 100 mM Tris-HCl/100 mM EDTA plus with 5 μl RNase A (10 mg/ml stock).

**Quantitative real-time PCR**. Approximately 100 flies were transferred to a 15-ml tube chilled on liquid nitrogen, decapitated by vigorously vortexing the tube containing the flies. The fly heads were then separated from the bodies and other parts by using metal sieves (pore size #25 and #40). Total RNA was extracted from head of flies using a TRIzol reagent (Invitrogen Corp., Carlsbad, CA), according to the manufacturer's instructions. We used FastStart Universal SYBR Green Master /ROX qPCR Master Mix (Thermo Fisher Scientific Inc., Waltham, MA USA) to conduct RT-PCR. RP49 was used as control for normalization. The primers used were as follows: RP49 (forward, 5′-CACACCAAATCTTACAAAATGTGTGA-3′; reverse: AATCCGGCCTTGCACATG), *Dsk* (forward, 5′-CCGATCCCAGCGCA GACGAC-3′; reverse: 5′-TGGCACTCTGCGACCGAAGC-3′) and *CCKLR-17D3* (forward, 5′-ACGCGTACCCTGTACGTAGG-3′; reverse:

GGTCTCGTTGTCAAGGTGGT). These primer sets were used for RT-PCR in Supplementary Fig. 7.

**Generation of the *DskGAL4* (attP2).** To make the *DskGAL4* construct, a 1.1-kb sequence upstream of the *Dsk* gene was PCR amplified from wild-type Canton-S flies using primers 5′-ACGACGTCAAGCTTATGGTCGGTCTCACCGTCA-CACTGT-3′ and 5′-TAGGTACCATGCTTTACTGTGCCCTTGGCAGA-3′ (the added restriction sites were underlined), and was cloned into pBPGUw[73] (a gift from Gerald Rubin, Addgene #17575) between the AatII and the KpnI sites.

**Generation of the *DskLexA* (attP2).** To make the *DskLexA* construct, the same 1.1-kb sequence upstream of the *Dsk* gene (digested with HindIII/KpnI), together with the coding sequence of *LexA::p65Uw* (excised with KpnI and XbaI) from pBPLexA::p65Uw[73] (a gift from Gerald Rubin, Addgene #26230) was cloned into pJFRC-MUH[74] (a gift from Gerald Rubin, Addgene #26213) between the HindIII and the XbaI sites.

**Generation of the *DskFlp* (attP2).** To make the *DskFlp* construct, the same 1.1-kb sequence upstream of *Dsk* gene (digested with HindIII/KpnI), together with the coding sequence of the Flp recombinase, was cloned into pJFRC-MUH[74] (a gift from Gerald Rubin, Addgene #26213) between the AatII and the XbaI sites. All DNA constructs were verified by sequencing, and were integrated at the attP2 site on the third chromosome with PhiC31-mediated transgenesis.

**Generation of the *Dsk* mutants.** Two gRNAs (gRNA1: 5′-CATTCTCTC-TATTCGGGGAC-3′; gRNA2: 5′-GACTACGGTCACATGCGTTT-3′) against *Dsk* were inserted into pCFD4[48] (a gift from Simon Bullock, Addgene #49411) as previously described. The *Dsk-gRNA* construct was then integrated at the attP40 site on the second chromosome. The *Dsk-gRNA* fly was crossed with *vas-Cas9* to screen for *Dsk* mutation as previously described. Using this method, we obtained two *Dsk* mutants (*ΔDsk*1 and *ΔDsk*2, Fig. 6a). The following primers were used to confirm *Dsk* mutations:

forward: 5′-CAGTAAAGCATGGGACCTAGAAGCTGT-3′;
reverse: 5′-TGTGTGCTCGATTAATTTCTATGTACA-3′.

**Generation of the *CCKLR-17D3* mutants.** Using the same method that generates the above *Dsk* mutants, we obtained the *Δ17D3* mutant by using the following two gRNAs (gRNA1: 5′-GGTCATCCGGGATGTTCAAC-3′; gRNA2: 5′-TCAA-CAGTCCTCAGCTCTAA-3′) against *CCKLR-17D3*. Candidates of *Δ17D3* were characterized by the loss of DNA band in the deleted areas by PCR on the genomic DNA, as shown in Fig. 7a. Primer sequences used for regions 1–5 in Fig. 7b are as follows:

Region 1: 5′-GCAAACACATAACGAGCCGAG-3′ and 5′-TATTGAAACGG CGACGCTTGC-3′.
Region 2: 5′-GGGATGTTCAACTACGAGGAG-3′ and 5′-GCAGAGGAACT CGCCAAAGAT-3′
Region 3: 5′-CGCTACTACGCGATATGCCAT-3′′ and 5′-ATTATAGACTG CGGTGGCGGT-3′.
Region 4: 5′-ACCACCTTGACAACGAGACCA-3′ and 5′-TTGGTGTTGGCA CTCGCATAG-3′.
Region 5: 5′-ATCAACGAGATGCGGTGTAAA-3′ and 5′-GCTTGTGCTCCA CTCAACTGT-3′

Primer sequences used for amplifying *CCKLR-17D3* or *CG6891* cDNA in Fig. 7c are as follows:

*CCKLR-17D3* cDNA: 5′-GGTCAAGATGCTGTTCGTCC-3′ and 5′-GGCGTT CATGAAGCAGTAGA-3′
*CG6891* cDNA: 5′-TGGTGGAGAGCAAGCCGAGAA-3′ and 5′-GGATGCGT ATGTAGCCAAAGG-3′

**Generation of the *17D3^GAL4* knock-in line.** *17D3^GAL4* was generated by replacement of the first *CCKLR-17D3* coding exon with *GAL4::p65* (Fig. 7f). Firstly, two gRNAs (gRNA1: 5′-CCGCAACGGGACATGTCAGG-3′; gRNA2: 5′-CACGG-CATGCCATTAGGGT-3′) against *CCKLR-17D3* were inserted into pCFD4 as previously described[48]. Secondly, we fused *GAL4::p65* into 5′ multiple cloning site of pHD-DsRed (a gift from Kate O'Connor-Giles, Addgene #51434) between the EcoRI and the NdeI sites. Then, each homologous arm was subcloned into the pHD-DsRed vector. The modified pCFD4 and pHD-DsRed plasmids were injected into the embryo of *vas-Cas9* flies. The correct insertion was confirmed by 3xP3-DsRed screening and direct sequencing.

**Generation of the Cas9 activating lines.** In order to enhance expression of *Dsk*, *CCKLR-17D3*, and *CCKLR-17D1*, an effective and convenient targeting activator system, flySAM, was applied[53]. The primers for sgRNA were annealed and ligated with the flySAM digested with BbsI, and the resulting constructs were injected into y sc v nanos-integrase; attP40 embryos following standard injection procedures[49,57]. The following are the sgRNA primers for the transgenic activation lines used in this study:

*Dsk* forward: 5′-ttcgGCCCAGCGCCCTAATACAGA-3′
*Dsk* reverse: 5′-aaacTCTGTATTAGGGCGCTGGGC-3′
*CCKLR-17D3* forward: 5′-ttcgCTCTGGCACTCAAGTGCCGT-3′
*CCKLR-17D3* reverse: 5′-aaacACGGCACTTGAGTGCCAGAG-3′
*CCKLR-17D1* forward: 5′-ttcgTTACCATCACTGAATCGTCG-3′
*CCKLR-17D1* reverse: 5′-aaacCGACGATTCAGTGATGGTAA-3′

**DSK antibody.** Rabbit anti-DSK antibody was generated by using the peptide N′-FDDYGHMRFC-C′ that corresponds to the predicted DSK-1 and DSK-2 peptides as antigen, and used throughout except for Supplementary Fig. 1, in which the antibody was obtained from Dr. David Petzel and described previously[75]. The antiserum was purified and used at 1:100 dilution. The antigen peptide synthesis and antiserum production were performed by GenScript Corp. (Nanjing, China).

**Male courtship assay.** For courtship assay, 4–8 days old wild-type virgin females were loaded individually into round 2-layer chambers (diameter: 1 cm; height: 3 mm per layer) as courtship targets, and 4–6 days old tester males were then gently aspirated into the chambers and separated from target females by a plastic transparent film until courtship test for 10 min. Inter-male courtship is assayed the same as above but using two males of the same genotype. Courtship index (CI), which is the percentage of observation time a male fly performs courtship (e.g., chasing, wing extension, circling, copulation), was used to measure courtship, and scored manually using the LifeSongX software.

**Ejaculation assay.** For single fly ejaculation experiments, individual males were anesthetized with $CO_2$ and then glued to a glass coverslip. After a 1-h recovery period in a humidified chamber, flies were recorded at 30 °C for 30 min and checked for ejaculation (Table S1).

**Female receptivity assay.** Four to eight days old virgin female flies and 4–6 days old wild-type virgin male flies were gently aspirated or iced singly into two layers of the round courtship chambers (diameter: 1 cm; height: 3 mm per layer) respectively and separated by a film between the layers. After about one hour's acclimation, the film was removed to allow the paired flies to get in contact, and courtship was recorded by camera for 30 min. Receptivity was measured every two minutes as the cumulative percentage of females engaging in copulation.

***dTrpA1* activation.** Flies were maintained at 22 °C, cold anesthetized and loaded into behavioral chambers, and allowed to recover for at least 30 min at 22 °C. Chambers containing flies were then placed at the appropriate control (22 °C) or experimental temperatures (30 °C) for 30 min before behavioral tests.

***CsChrimson* activation.** For all the *CsChrimson* experiments, crosses were set up on standard fly food in vials that covered by aluminum foil to protect from light. Tester flies were collected immediately after eclosion and reared in groups of 15–20 on 0.2 mM retinal (116–31–4, Sigma-Aldrich) food in vials that covered by aluminum foil for 4–6 days before courtship test. Courtship was performed as above described but either in dark (as control) or under red LED light stimulation (620 nm, 0.03 mW/mm2, Vanch Technology, Shanghai, China). Light intensity was measured by placing an optical power meter (PS-310 V2, Gentec, Canada) nearby the location of chambers. Fly behavior was recorded by a Stingray camera equipped with an infrared filter under 860-nm IR LED illumination (Vanch Technology, Shanghai, China).

**Sleep test.** Individual 2–4 day old males were placed in locomotor activity monitor tubes (DAM2, TriKinetics Inc) with fly food, and were entrained in 22 °C 12 h:12 h light: dark conditions for at least 2 days before sleep test. For sleep test in Fig. 4b, 1-day sleep data were recorded at 22 °C as baseline, then flies were shifted to 30 °C for two days, and returned to 22 °C for one day. Sleep was analyzed using custom designed Matlab software[76]. Change in total sleep (e.g. Figure 4c) is the percentage of sleep change in the first day of temperature shift (30 °C) compared to baseline sleep at 22 °C.

**Spontaneous locomotion assay.** Flies were transferred singly into round wells of 2-cm diameter and 3-mm height covered by regular food, and recorded for 24 h starting from 9 am under constant light condition. The average walking velocity during the 24-h recording was quantified using the ZebraLab software system (ViewPoint Life Sciences, Montreal, Quebec, Canada) as previously used[33,44].

**Visual-induced locomotion assay.** We used a LED arena to evaluate optomotor response and visual-induced walking activity of flies. In brief, individual wing-cut male flies were allowed to walk for 6 min on a circular platform, 86 mm in diameter, surrounded by a water-filled moat to prevent escape. The moat was surrounded by a panoramic LED display, 290 mm in diameter and 345 mm in height. The LED display was a cylinder of evenly distributed 128 (row) × 32 (column) LED units and was computer-controlled with LED Studio software (Shenzhen Sinorad Medical Electronics, Shenzhen, China). The refresh rate of the LED panels was 400

Hz. A camera (WV-BP330, Panasonic System Networks, Suzhou, China) directly above the arena was connected to a computer to record the fly's walking track at a rate of 12 frames per second, and position coordinates of the fly in each frame were calculated using Limelight software (Coulbourn Instruments, Whitehall, PA, USA).

The flies was presented with VisionEgg[77] generated horizontally moving gratings (spatial frequency:12/128; contrast: 100; temporal frequency: 3 Hz; orientation: 180°) first with 3 min clockwise rotation and then 3 min counter-clockwise rotation. Optomotor performance index was measure by the equation $PI = (t_{syn}-t_{anti})/(t_{syn}-t_{anti})$, in which $t_{syn}$ means the time the fly's turning was in the same direction as the grating movement while $t_{anti}$ means the time the fly's turning was in the opposite direction as the grating movement. The average walking velocity during the 6-min test was also calculated to reflect visual-induced walking activity.

**Feeding assay.** Two feeding assays were used in this study. First, the capillary feeding assay (CAFE) was conducted as previously reported[78] with slight modifications. Briefly, male flies were placed into 1.5-ml Eppendorf microcentrifuge tubes with 200 μl 1% agar for water consumption, each with an inserted calibrated pipets (5 μl, catalog no. 53432-706; VWR, West Chester, PA) with 2.5% sucrose, 2.5% yeast extract, and 0.1% propionic acid. Five food filled capillaries were inserted as controls in identical tubes without flies. The final consumption of food was determined as the decreased food level minus the average decrease in control capillaries. Daily consumption was measured over four consecutive days. Second, feeding was also assayed on food with blue dye. In brief, flies were starved for 24 h on 1% aqueous agarose at 22 °C. Then they were moved to 30 °C for 30 min for dTrpA1 activation. Thereafter, they were transferred to 1% FD&C Blue 1 (Sigma-Aldrich) colored food (2.5% sucrose, 2.5% yeast extract, and 0.5% agarose) for 15 min at 30 °C allowing feeding. The flies were observed under a light microscope and scored for blue color in their abdomens. To quantify the food intake of the flies more accurately, the absorbance of the ingested blue dye was measured as previously described[79]. 20 flies were decapitated, the bodies were collected in 1.5 ml microcentrifuge tubes, homogenized in 500 μl distilled water, and centrifuged for 1 min at 12,000g. Three 100 μl samples of supernatant from each probe were taken, and absorbance of blue dye was quantified using a 96-well microplate spectrophotometer at 630 nm.

**Tissue dissection, staining, and imaging.** We dissected brains and ventral nerve cords of 4–6 days old male or female flies in Schneider's insect medium (S2) and fixed them in 4% paraformaldehyde (PFA) in phosphate buffered saline (PBS) for 30 min at room temperature. After four washes of 15 min (4 × 15 min) in PAT (0.5% Triton X-100, 0.5% bovine serum albumin in PBS), tissues are blocked in 3% normal goat serum (NGS) for 60 min, then incubated in primary antibodies diluted in 3% NGS for circa 24 h at 4 °C, then washed four times (4 × 15 min) in PAT and incubated in secondary antibodies diluted in 3% NGS for circa 24 h at 4 °C. Tissues are then washed four times (4 × 15 min) in PAT and mounted in Vectorshield (Vector Laboratories) for imaging. Primary antibodies used: rabbit anti-GFP (Invitrogen A11122, 1:1000), mouse anti-Bruchpilot (Developmental Studies Hybridoma Bank nc82, 1:30), rabbit anti-DSK (see antibody generation section, 1:100). Secondary antibodies used: goat anti-mouse IgG conjugated to Alexa 488 (1:500) or Alexa 555 (1:500) and goat anti-rabbit IgG conjugated to Alexa 488 (1:500) or Alexa 555 (1:500) (Molecular Probes). Samples were imaged at ×20 magnification on Zeiss 700 confocal microscopes, and processed with ImageJ.

To quantify DSK level in Dsk neurons, we co-stained the brains with DSK antibody and GFP antibody as control signals (Supplementary Fig. 6a). The samples of the same experiments were processed in parallel and using the same solution and imaged with the same laser power and scanning settings. With the imaged data, we got "Sum Slices" Z-projection of the substacks encompassing the cell bodies of eight MP1 and MP3 Dsk neurons and select the soma areas (or whole brain) to measure fluorescence intensity ($F_{DSK}$ and $F_{ctrl}$), then select a small region without signal as the background fluorescence ($B_{DSK}$ and $B_{ctrl}$) in both channels using ImageJ. Then we got the real signal in both channels by subtracting the background fluorescence from total signal respectively, and got the relative fluorescence of DSK as the ratio of DSK signal to the control signal (($F_{DSK} - B_{DSK}$)/($F_{ctrl} - B_{ctrl}$)).

**Experience-dependent syb-GRASP quantification.** For visualization of synaptic contacts, syb-GRASP method was utilized as previously described[46]. Briefly, Males were collected after eclosion under three housing conditions: singly housed, male–male group-housed (11 males) and male–female group-housed (1 male and 10 wild-type virgin females). Flies were transferred to vials with fresh food every three days, and males were dissected and imaged at the age of two weeks. Monoclonal mouse anti-GFP antibody (G6539, Sigma-Aldrich) was used to visualize reconstituted GFP.

All samples were stained and imaged under the same condition. Monochrome images were rendered to emphasize differences in intensity. For each region of interest (ROI), a maximum z-projection of a fixed number of image stacks with GRASP signals was created, and average fluorescence intensity was calculated for each sample.

**Stochastic labeling and manipulation of Dsk neurons.** Virgin hs-Flp;; DskGAL4 females were crossed with UAS > stop > CsChrimson-mVenus males at 25 °C on retinal food in vials that covered by aluminum foil. Heat shock was performed using 37 °C water bath for 10 min when early pupae begin to form. Males were collected after eclosion and fed in groups of 15–20 on retinal food in vials that covered by aluminum foil. Four to six days old males were assayed for courtship using wild-type females as targets under constant red light stimulation as above mentioned. One hundred and thirty two males were recorded for courtship for 30 min and analyzed into three categories: (1) 78 males did not initiate courtship (non-courters); (2) 19 males showed lower level of courtship and did not copulate with females; and (3) 35 males courted and successfully mated with females (courters). We then dissect the first (non-courters) and third (courters) categories after courtship assay and analysis, and image CsChrimson-mVenus expression using rabbit anti-GFP and secondary antibodies as above mentioned. 73 non-courters and 30 courters were successfully imaged and then analyzed for their expression in MP1a, MP1b, and MP3 neurons.

**Brain image registration.** The standard brain used in this study is described previously[58]. Confocal images for a single MP neuron (MP1a or MP1b, this study) and P1 neurons (P1-splitGAL4) were registered onto this standard brain with a Fiji graphical user interface (GUI) as described previously[80].

**Statistics.** Experimental flies and genetic controls were tested at the same condition, and data are collected from at least two independent experiments. Statistical analysis is performed using GraphPad Prism and indicated inside each figure legend. Data presented in this study were first verified for normal distribution by D'Agostino–Pearson normality test. If normally distributed, Student's t test is used for pairwise comparisons, and one-way ANOVA is used for comparisons among multiple groups, followed by Tukey's multiple comparisons. If not normally distributed, Mann–Whitney U test is used for pairwise comparisons, and Kruskal–Wallis test is used for comparisons among multiple groups, followed by Dunn's multiple comparisons.

**Reporting summary.** Further information on research design is available in the Nature Research Reporting Summary linked to this article.

## Data availability

The source data underlying Figs. 1, 2, 3, 4, 5, 6, 7, and Supplementary Figs 2, 3, 4, 5, 6, 7, 8, 8, 9, 10 are provided as a Source Data file. Sequences of ΔDsk1, ΔDsk2 and Δ17D3 that generated in this study were deposited to GeneBank (accession numbers are MN257953, MN257954 and MN257955 respectively). All relevant data are available from the corresponding author upon reasonable request.

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

## Acknowledgements

The authors thank the Tsinghua Fly Center, Janelia Research Campus and Bloomington Stock Center for fly stocks. We thank Dr. Li Liu and Deliang Yuan for help with the visual tracking assay. This work was supported by the National Natural Science Foundation of China (31622028, 31571093 to Y.P., 31772205 to S.W., and 6531000063 to C.G.), the China Postdoctoral Science Foundation (2015M581692 to S.W.), the Natural Science Foundation from Jiangsu Province (BK20160025 to Y.P.), the Jiangsu Innovation and Entrepreneurship Team Program, and the Fundamental Research Funds for the Central Universities (2242018K41056 and 2242018K3DN07).

## Author contributions

Conceptualization, S.W., C.G., S.D.L. and Y.P.; Methodology, J.C., C.H., H.Q. P.P. and Y.L.; Investigation, S.W., C.G., H.Z., M.S., Q.P., S.D.L. and Y.P.; Writing—Original Draft, Y.P.; Writing—Review & Editing, S.W., C.G., Q.P. and Y.P.; Funding Acquisition, S.W., C.G. and Y.P.; Supervision, Y.P.

## Competing interests

The authors declare no competing interests.
