## [Peer Review File · Nature Communications]

Reviewers' comments:

Reviewer #1 (Remarks to the Author):

This study demonstrates the importance of four pairs of large peptidergic neurons in the *Drosophila* brain in inhibition of male sexual behavior. These neurons (MP1 and MP3 neurons) produce the neuropeptides sulfakinin 1 and 2 (DSK1 and 2) and arborize in wide areas of the brain. Furthermore they express the male-specific form of fruitless (FruM) and are thus part of a large population of neurons important in courtship regulation. The role of DSK in inhibition of courtship was detected in a screen of 32 GAL4 lines representing neuropeptidergic neurons. Using specific Dsk-GAL4 drivers the authors performed thermogenetic and optogenetic experiments to demonstrate the role in courtship. By intersectional techniques it was shown that the MP1a and b and MP3 neurons (with different morphologies) were sufficient for courtship inhibition and that one of the two known Dsk receptors, CCKLR-17D3, is also needed. These MP neurons were shown by various genetic techniques to be both upstream and downstream of FruM expressing P1 neurons, known to be neurons integrating chemosensory cues and stimulate male sexual arousal. Direct synaptic contacts were only demonstrated from P1 neurons to MP neurons, but it was proposed that non-synaptic (paracrine) signaling could operate for MP to P1 neurons. The authors also provide evidence that DSK peptides in the MP neurons are required (as opposed to some other neurotransmitter in these neurons) by peptide knockdown, and by conditional over-expression. The CCKLR-17D3 was shown to be expressed (GAL4 driver) in FruM neurons, including P1 neurons, and receptor mutants phenocopied peptide/MP manipulations. Conditional receptor overexpression decreased male courtship. The MP neurons also inhibit female sexual receptivity, but by a different circuitry (since P1 neurons are lacking). Fig. 7 nicely summarizes the findings.

Overall it is a nice and thorough study that clearly demonstrates the role of DSK and the MP neurons in inhibition of male courtship. It is not entirely clear from the paper what conditions (internal or external cues) or neurons that drive the MP neurons (although some indications in Suppl Fig 8). Since DSK has been indicated in several other functions (aggression, activity, satiety and so on), it would be of interest to know what other inputs the MP neurons have.

There are a few studies on neurons producing another peptide, SIFamide (SIFa), that are not cited in the present study and that could serve as guide. SIFa also inhibits male reproductive behavior (Terhaz et al. 2007; Sellami and Veenstra 2015) and has additionally been shown to stimulate appetite and feeding behavior and inhibit sleep. The SIFa neurons receive satiety (MIP peptide) and hunger (Hugin-PK peptide) signals from peptidergic neurons and modulate olfactory and gustatory circuits, but also affect locomotor activity and sleep, as well as male reproductive behavior by acting on FruM neurons (Martelli et al., 2017, Sellami and Veenstra 2015). These SIFa neurons may receive additional inputs from neurons producing corazonin, insulin-like peptides, DSK and sNPF (as suggested by GRASP technique). Thus, SIFa neurons act context-dependently to regulate feeding, sleep and mating. Would be interesting to determine possible functional connections/interactions between SIFa neurons and DSK neurons in reproductive behavior (the present authors also found roles of corazonin and MIP in their screen). At least the authors should discuss the findings in these papers (see also a recent paper where this is discussed in a more integrated form – Fig. 3 in: <https://doi.org/10.7287/peerj.preprints.27531v1> BUT no need to cite that paper!). Since DSK seems to be part of satiety-hunger circuits, that internal state may be critical also in the DSK role in mating (as is the case for SIFa neurons) indications in Suppl Fig 8.

Specific comments

1. To line 67-69 and 74-75: There is an inhibitory counterpart, namely SIFa (see above). I think this needs to be mentioned. The SIFa neurons relay internal states (hunger-satiety).
2. Line 88 and Fig. 1: I am curious why SIFa-GAL4 was not used in the screen as a positive control (see above). Sorry about nagging about this peptide!
3. A comment to Fig. 2. The antiserum to DSK labels far more neurons than the Dsk-GAL4 expression. Can the authors comment on which labeling pattern is likely to be correct? Generating

an antiserum to an FMRFamide containing sequence could result in some cross reactivity to other peptides containing (FM)RFamide, but of course GAL4 expression can also be variable. From previous data in many insect species, it is likely that there are no DSK-expressing neuronal cell bodies in the thoracic-abdominal ganglia, suggesting that the antiserum cross-reacts somewhat with other peptides.

4. Line 307: I do not understand the reasoning behind the statement "These results indicate that DSK-2, but not DSK-0 or DSK-1, is responsible for courtship inhibition." The two mutants delete either all DSKs or DSK2, so the latter mutant is sufficient for an inhibition, but did the authors test whether levels of DSK1 were also affected in that mutant for other reasons than its coding region missing. I find it really remarkable that DSK2 alone would activate the CCKL receptor, given that the DSK1 and 2 are so similar.

5. Line 320: The authors used conditional acute overexpression of Dsk. How do the authors explain the inhibition of courtship in this case? Overexpression of a peptide in wild type background probably leads to increased production and storage of DSKs, BUT how could that lead to increased release during the behavior? Release is regulated by neuron depolarization and it is likely that a depolarization results in the same amount of release no matter how much peptide the neurons contains. Is there any other explanation? Presumably neuronal activity is the same as in control flies; only the DSK content in MP neurons is different. Overexpression of the CCKLR (line 336-) is a different matter since more receptors probably means greater signal transfer.

6. Suppl. Fig. 8 is not utilized in the Results part. Why is that? It summarizes experiments performed and should be referred to in the Results section instead of the Discussion.

7. The results in Suppl. Fig. 8 are interesting in that they may suggest that the receptor expression is indeed state dependent, including state of feeding. Maybe also inputs to MP neurons are state dependent (orexigenic, anorexigenic, sleep-wake). Monitoring Ca activity in MP neurons would be a possible way to determine what regulates their activity (I do not request such experiments to be done here).

8. To Line 416: Redundant inhibitory signals include SIFa neurons signaling to FruM neurons.

9. A comment to figures with images of brains/neurons: Most of these images (at least in my version) are extremely small and it is hard to see the neuron branching (only on line/screen with zoom is it possible, not in a print-out). It seems like the higher impact factor a journal has the smaller the images are made! This is a pity since the anatomy is rather important in this paper. Especially double labeling and intersectional labeling is hard to resolve.

References in text above:

Martelli C, Pech U, Kobbenbring S, Pauls D, Bahl B, Sommer MV, Pooryasin A, Barth J, Arias CWP, Vassiliou C, et al.: SIFamide Translates Hunger Signals into Appetitive and Feeding Behavior in *Drosophila*. *Cell Reports* 2017, 20:464-478.

Terhzaz S, Rosay P, Goodwin SF, Veenstra JA: The neuropeptide SIFamide modulates sexual behavior in *Drosophila*. *Biochem Biophys Res Commun* 2007, 352:305-310.

Minor edits

1. The sentence on line 25-26 is awkward: "...to understand THEM" - what is them referring to?

2. Line 29 (see also Line 66): P1 neurons a "command center". The P1 neurons are part of a command circuit.

3. Line 34: Is it really clear that Fru neurons regulate ALL aspects of sexual behaviors. This is a strong statement.

4. Line 613: what is positively and negatively purified?

Sellami A, Veenstra JA: SIFamide acts on fruitless neurons to modulate sexual behavior in *Drosophila melanogaster*. *Peptides* 2015, 74:50-56.

Reviewer #2 (Remarks to the Author):

This paper reports on the discovery of a few fru-positive interneurons (MP1) that inhibit activities in P1 neurons, the primary decision-making center for male courtship behavior. The newly discovered courtship-inhibiting neurons likely release the neuropeptide Drosulfakinin-2 (DSK-2) to suppress activities in P1 neurons, which nonetheless turned out to be presynaptic to the Dsk-positive interneurons. Remarkably, activity-dependent GRASP signals at the P1 – MP1 neuronal connections appeared to be enhanced by male – male social interactions, while other neural inputs impinging on MP1 were differently affected by rearing conditions. Overall, the experiments were carefully conducted using an impressive array of newly-emerged techniques, which well substantiated the conclusion that MP1 represents a state-dependent negative regulator of courtship activities in flies. This work is worth publishing after the clarification of certain ambiguity.

1. The authors emphasize the importance of neural integration of internal states and external stimuli. This raises a question what external stimuli may provoke MP1 activities. Social experience must be converted into some neural codes, likely initiated by sensory cells and ultimately impacts MP1. Although the identification of such an input pathway to MP1 is beyond the scope of this paper, the authors are encouraged to discuss which sensory modality is involved and how the social cue is conveyed to MP1.

2. "FRU[M] is both necessary and sufficient" (line 52) is not warranted and thus to be removed (see more details: Yoshihara and Yoshihara, 2018 <https://www.ncbi.nlm.nih.gov/pubmed/29757057>).

3. The authors found a prolonged inhibitory effect after transient activation of MP1 on male courtship (lines 123-129). How could MP1 activation give rise to a persistent effect after the cessation of optogenetic stimulations? What would be the biological advantage of such long-lasting suppression of male courtship? These issues merit discussion.

4. Potential connections from different sets of dsx-positive neurons to MP neurons appear to be enhanced under different social contexts (lines 272-283; Fig. 4j-n). What are the possible neural groups that are responsible for altered activities in ROIs 1-3?

5. The authors claim that immunoreactivity to the anti-DSK antibody at presynaptic terminals is decreased after TrpA1-mediated activation of MP neurons. However, the data shown in Supplementary Fig. 6b indicate that the anti-DSK staining intensity decreased throughout a neuron, not just at presynaptic terminals. In this case, the way to describe the observed changes is misleading. If presynaptic terminals were more sensitive to forced activation than other neural parts, the authors should show higher magnification views of a neuron stained for DSK, making it possible to compare the staining intensity among different portions of the cell.

6. Functional GRASP data suggested that P1 synapses on to DSKergic MP1, whereas the DSK receptor CCKLR-17D3 was expressed in P1. In fact, the authors seem to imply bidirectional P1 – MP neuron communications (lines 475-461). What mechanism could mediate the MP-to-P1 information flow? Do you suggest an involvement of systemic DSK or any retrograde actions across synapses? A schematic illustrating hypothetical neural and humoral pathways that may contribute to P1-MP interactions for regulating courtship and sleep would help readers understand the provisional network underlying behavioral choice switch.

7. Based on functional GRASP assays, the authors argue that male-to-male interactions potentiate

synaptic transmission from P1 to MP neurons (lines 269-272). However, one cannot exclude the possibility that stronger GRASP signals observed at P1-to-MP synapses were a secondary consequence of the general elevation of P1 activity levels upon male-to-male interactions, not reflecting any augmentation specific to these synapses. The authors' observation that many connections between different sets of dsx-positive neurons and MP neurons were boosted by social experience seems to be consistent with this idea. Kohatsu and Yamamoto (2015) previously reported that responsiveness to moving visual targets of putative P1 neurons is susceptible to housing conditions, at least in fru mutant males. Do you think that enhanced P1-to-MP transmission ultimately suppresses male-directed courtship in these males? The discussion needs to be expanded to include possible mechanisms for enhanced GRASP signals and behavioral roles of such enhanced transmission.

Reviewer #3 (Remarks to the Author):

In this study, Wu and colleagues explore the intriguing question of how internal states interact and are regulated by examining how peptidergic systems modulate *Drosophila* courtship. They demonstrate that activation of a small number of Fru+ drosulfakinin-expressing neurons in the medial protocerebrum strongly suppresses courtship and suggest that this effect appears to be mediated by drosulfakinin acting via one of its cognate receptors to modulate the activity of male-specific P1 neurons that serve to initiate courtship behavior. The authors show that co-activation of these Dsk+ neurons and P1 neurons lead to intermediate phenotypes in both courtship and sleep assays, in accord with an antagonistic interaction between these two populations. Broadly, I found the results interesting and convincing, and appreciate the general thrust of the manuscript. One limitation of this work is that the functional logic of the Dsk modulation is largely derived from behavioral manipulations, many of which rely on artificial activation of Dsk and/or P1 neurons. If the authors could more clearly elucidate the interactions and chain-of-command in the proposed circuitry, this work would certainly represent an advance in thinking about the regulation of motivational states.

One major concern in the current manuscript is that the natural role of Dsk neurons in natural courtship is not clear, nor exactly over what time scales they affect downstream circuitry. While the authors nicely demonstrate that acute activation of these neurons strongly reduce courtship, and that this effect is dependent on one of the Dsk receptors, I remain a little unclear from the data presented about when these neurons would be normally activated to regulate courtship in this manner. This concern is more pronounced by the lack of behavioral evidence that silencing Dsk neurons or knocking out Dsk has an impact on normal courtship behavior. A related point is that the authors should discuss the time scale on which these neurons are proposed to regulate behavior—acutely, over minutes (as Fig. 1 would suggest) in response to changes in sensory input, or over hours or days (as Fig. 4 would suggest) or the lifetime of the fly (as Fig. S8 would suggest) in responses to past experience or changes in their mating drive?

The logic behind P1 neurons synapsing directly onto Dsk neurons should also be clarified. Intuitively, if the authors believe that the Dsk neurons are acting via P1 neurons, it would seem that P1 activation would simultaneously drive and suppress courtship. This again raises a question regarding the time-scale over which these neurons function, but also raises the possibility that Dsk suppression could be mediated by a distinct downstream target. Indeed, GRASP signals are much greater when all dsx+ neurons are included suggestive that Dsk signaling may just as readily act to suppress courtship via a common target of both P1 and Dsk+ neurons. The authors should experimentally address whether Dsk signaling onto P1 neurons is driving courtship suppression, perhaps by knocking down the CCKLR receptors in the neurons targeted by the split-P1 GAL4 line. The authors rule out the possibility of Dsk neurons driving a competing behavior (which would parsimoniously explain the results) on the basis of five arguments (lines 426-432). It is not fully clear to me, however, that this conclusion logically flows from these arguments nor other data. For

example, there appears to be a systemic decrease in the velocity of DskGal4 > dTrpA1 neurons relative to controls (Fig. 3g-j; the statistical test should account for repeated measures). The monomorphic nature of many of these neurons would further strengthen this hypothesis. The authors should discuss the possibility that Dsk neurons drive a competing behavior, phenotypically shunting courtship.

Lastly, there appears to be a quite strong suppression of courtship in the control empty Gal4 line in Fig. 1d. This is a very good control, but the authors should discuss this effect, and their choice to not use it as a control for the TrpA1 experiments in Fig. 3.

Our responses to comments are immediately after each comment, and in blue:

Reviewer #1 (Remarks to the Author):

This study demonstrates the importance of four pairs of large peptidergic neurons in the *Drosophila* brain in inhibition of male sexual behavior. These neurons (MP1 and MP3 neurons) produce the neuropeptides sulfakinin 1 and 2 (DSK1 and 2) and arborize in wide areas of the brain. Furthermore they express the male-specific form of fruitless (FruM) and are thus part of a large population of neurons important in courtship regulation. The role of DSK in inhibition of courtship was detected in a screen of 32 GAL4 lines representing neuropeptidergic neurons. Using specific Dsk-GAL4 drivers the authors performed thermogenetic and optogenetic experiments to demonstrate the role in courtship. By intersectional techniques it was shown that the MP1a and b and MP3 neurons (with different morphologies) were sufficient for courtship inhibition and that one of the two known Dsk receptors, CCKLR-17D3, is also needed. These MP neurons were shown by various genetic techniques to be both upstream and downstream of FruM expressing P1 neurons, known to be neurons integrating chemosensory cues and stimulate male sexual arousal. Direct synaptic contacts were only demonstrated from P1 neurons to MP neurons, but it was proposed that non-synaptic (paracrine) signaling could operate for MP to P1 neurons. The authors also provide evidence that DSK peptides in the MP neurons are required (as opposed to some other neurotransmitter in these neurons) by peptide knockdown, and by conditional over-expression. The CCKLR-17D3 was shown to be expressed (GAL4 driver) in FruM neurons, including P1 neurons, and receptor mutants phenocopied peptide/MP manipulations. Conditional receptor overexpression decreased male courtship. The MP neurons also inhibit female sexual receptivity, but by a different circuitry (since P1 neurons are lacking). Fig. 7 nicely summarizes the findings. Overall it is a nice and thorough study that clearly demonstrates the role of DSK and the MP neurons in inhibition of male courtship. It is not entirely clear from the paper what conditions (internal or external cues) or neurons that drive the MP neurons (although some indications in Suppl Fig 8). Since DSK has been indicated in several other functions (aggression, activity, satiety and so on), it would be of interest to know what other inputs the MP neurons have.

There are a few studies on neurons producing another peptide, SIFamide (SIFa), that are not cited in the present study and that could serve as guide. SIFa also inhibits male reproductive behavior (Terhaz et al. 2007; Sellami and Veenstra 2015) and has additionally been shown to stimulate appetite and feeding behavior and inhibit sleep. The SIFa neurons receive satiety (MIP peptide) and hunger (Hugin-PK peptide) signals from peptidergic neurons and modulate olfactory and gustatory circuits, but also affect locomotor activity and sleep, as well as male reproductive behavior by acting on FruM neurons (Martelli et al., 2017, Sellami and Veenstra 2015). These SIFa neurons may receive additional inputs from neurons producing corazonin, insulin-like peptides, DSK and sNPF (as suggested by GRASP technique). Thus, SIFa neurons act context-dependently to regulate feeding, sleep and mating. Would be interesting to determine possible functional connections/interactions between SIFa neurons and DSK neurons in reproductive behavior (the present authors also found roles of corazonin and MIP in their screen). At least the authors should discuss the findings in these papers (see also a recent paper where this is discussed in a more integrated form – Fig. 3 in: <https://doi.org/10.7287/peerj.preprints.27531v1> BUT no

need to cite that paper!). Since DSK seems to be part of satiety-hunger circuits, that internal state may be critical also in the DSK role in mating (as is the case for SIFa neurons) indications in Suppl Fig 8.

We are gratified by the reviewer's acknowledgement of the significance and depth of our findings.

We thank the reviewer for pointing out the inhibitory role of SIFa in male courtship behavior. We previously did not test the role of SIFa-neurons in male courtship during initial behavioral screen, as all GAL4 lines we used were from Bloomington stock center, which does not have the SIFa-GAL4 line. In order to test whether SIFa-neurons play inhibitory role in regular male-female courtship, we requested two independently generated SIFa-GAL4 from Yi Rao's lab in Peking University, one is originally generated by Terhzaz et al., (2007), and the other recently published by Yi Rao's lab (Deng et al., 2019), and tested male-female courtship in males with these GAL4-labeled neurons activated. Activating these SIFa-GAL4 neurons using either driver (we have checked two GAL4s' expression and they are consistent with original publications) did not affect male-female courtship at all, indicating that SIFa-GAL4 neurons do not inhibit courtship towards virgin females. Note that previous findings only show that SIFa inhibits male-male courtship, but not male-female courtship (Terhzaz et al., 2007; Sellami and Veenstra, 2015). Thus, although SIFa inhibits inappropriate male-male courtship, it does not inhibit regular male-female courtship like DSK does.

Based on the above-mentioned new data and reasoning, we have now added the new data into Fig. 1b, and have made textual changes in both the introduction and discussion section to more accurately reflect the inhibitory regulation of male courtship by these neuropeptides. However, we are not able to dissect functional relationship between these two signaling due to both the limited scope of this study and the lack of sufficient genetic tools for SIFa signaling (*e.g.*, validated LexA, antibodies, mutants, receptor GAL4, receptor mutants).

Specific comments

1. To line 67-69 and 74-75: There is an inhibitory counterpart, namely SIFa (see above). I think this needs to be mentioned. The SIFa neurons relay internal states (hunger-satiety).

We have added the work of SIFa in the introduction, and also discussed different inhibitory function between SIFa and DSK in male courtship behavior.

2. Line 88 and Fig. 1: I am curious why SIFa-GAL4 was not used in the screen as a positive control (see above). Sorry about nagging about this peptide!

As stated above, we did not test the role of SIFa-neurons in male courtship during our initial screen, as we cannot find a SIFa-GAL4 line in Bloomington stock center, and also because SIFa only inhibits same-sex courtship but not male-female courtship. We now get two SIFa-GAL4 lines from Yi Rao's lab as their recent Neuron paper used both of these drivers (Deng et al., 2019).

3. A comment to Fig. 2. The antiserum to DSK labels far more neurons than the Dsk-GAL4 expression. Can the authors comment on which labeling pattern is likely to be correct? Generating an antiserum to an FMRFamide containing sequence could result

in some cross reactivity to other peptides containing (FM)RFamide, but of course GAL4 expression can also be variable. From previous data in many insect species, it is likely that there are no DSK-expressing neuronal cell bodies in the thoracic-abdominal ganglia, suggesting that the antiserum cross-reacts somewhat with other peptides.

Response: The DSK antibodies generated in this study as well as the one used by Nichols and Lim (1996) both faithfully labeled MP neurons (also labeled by our DskGAL4) in the fly brain, and a few neurons in the ventral nerve cord (namely as Tv and A8 neurons by Nichols and Lim, but did not labeled by our DskGAL4). We think it is possible that both DSK antibodies may cross-react with other peptides, thus those Tv and A8 neurons are not actually DSK-expressing. It is also possible that DskGAL4 only labels a part of Dsk-expressing neurons in the brain as we only used 1.1kb fragment upstream of the Dsk gene to generate the DskGAL4 line. We think generation of a knock-in GAL4 line into the locus of Dsk gene in the future may help to reveal whether those Tv and A8 neurons in the VNC are really Dsk-expressing neurons. We have made textual changes to more accurately describe expression of the *Dsk* gene.

4. Line 307: I do not understand the reasoning behind the statement “These results indicate that DSK-2, but not DSK-0 or DSK-1, is responsible for courtship inhibition.” The two mutants delete either all DSKs or DSK2, so the latter mutant is sufficient for an inhibition, but did the authors test whether levels of DSK1 were also affected in that mutant for other reasons than its coding region missing. I find it really remarkable that DSK2 alone would activate the CCKL receptor, given that the DSK1 and 2 are so similar.

We thank the reviewer for pointing out this issue, and indeed we do not have sufficient evidence to discriminate the role of DSK-1 and DSK-2. We have a mutant lacking all DSK peptides, and a mutant lacking only DSK-2, but we do not have a mutant lacking only DSK-1. What we found is that when Dsk neurons were activated, courtship inhibition requires DSK-2 peptides (as loss of DSK-2 results in restore of male courtship). Thus we choose not to compare the role of different DSK peptides, and accurately describe what we found and conclude as following: “These results indicate that DSK-2 is indispensable for courtship inhibition in males with Dsk neurons being activated, and further evidences are needed to reveal the role of DSK-1 in courtship inhibition”.

5. Line 320: The authors used conditional acute overexpression of Dsk. How do the authors explain the inhibition of courtship in this case? Overexpression of a peptide in wild type background probably leads to increased production and storage of DSKs, BUT how could that lead to increased release during the behavior? Release is regulated by neuron depolarization and it is likely that a depolarization results in the same amount of release no matter how much peptide the neurons contains. Is there any other explanation? Presumably neuronal activity is the same as in control flies; only the DSK content in MP neurons is different. Overexpression of the CCKLR (line 336-) is a different matter since more receptors probably means greater signal transfer.

We think overexpression of DSK may strengthen DSK-CCKLR signaling that acts on many *fru*-expressing neurons including P1 neurons (thus not only affect the excitability of the four pairs of MP neurons, but also many parts of the courtship circuitry) in males before courtship test, making this courtship circuit or a part of this circuit (e.g. P1 neurons) less responsive to virgin females, such that in the presence of virgin females, males displayed lower level of courtship. These experiments as well as how DSK peptides physiologically functions in MP neurons and how they communicate with P1 neurons to balance inhibitory and excitatory stimulation in male courtship behavior is of particular interest in our future study. Please also note that overexpression of neuropeptides affected many other behaviors, e.g., overexpression of Tk promoted male aggression (Asahina et al., 2014, Cell).

6. Suppl. Fig. 8 is not utilized in the Results part. Why is that? It summarizes experiments performed and should be referred to in the Results section instead of the Discussion.

We thank the reviewer for this suggestion, and we have now moved description of this data into the result section.

7. The results in Suppl. Fig. 8 are interesting in that they may suggest that the receptor expression is indeed state dependent, including state of feeding. Maybe also inputs to MP neurons are state dependent (orexigenic, anorexigenic, sleep-wake). Monitoring Ca activity in MP neurons would be a possible way to determine what regulates their activity (I do not request such experiments to be done here).

We thank the reviewer for this suggestion. Indeed, we are very interested in how DSK neurons are regulated by internal states and past experiences, which is actually a main focus in our future study. We believe further understanding of how DSK neurons integrate internal states and past experiences, and how they antagonize excitatory courtship center P1 neurons to determine a male's motivational state to court or not, will help us to understand motivation and decision-making in a more general way.

8. To Line 416: Redundant inhibitory signals include SIFa neurons signaling to FruM neurons.

We have re-wrote the sentence, and discussed the different roles of SIFa and DSK in courtship inhibition.

9. A comment to figures with images of brains/neurons: Most of these images (at least in my version) are extremely small and it is hard to see the neuron branching (only on line/screen with zoom is it possible, not in a print-out). It seems like the higher impact factor a journal has the smaller the images are made! This is a pity since the anatomy is rather important in this paper. Especially double labeling and intersectional labeling is hard to resolve.

We thank the reviewer for this suggestion, and made the neuronal labeling of single Dsk neurons (previously Fig. 2e, together with Fig. 2f) into a separate figure (now fig. 3), with much higher resolution so readers could easily see the morphology of single Dsk neurons in a printed version.

References in text above:

Martelli C, Pech U, Kobbenbring S, Pauls D, Bahl B, Sommer MV, Pooryasin A, Barth J, Arias CWP, Vassiliou C, et al.: SIFamide Translates Hunger Signals into Appetitive and Feeding Behavior in *Drosophila*. *Cell Reports* 2017, 20:464-478.

Terhzaz S, Rosay P, Goodwin SF, Veenstra JA: The neuropeptide SIFamide modulates sexual behavior in *Drosophila*. *Biochem Biophys Res Commun* 2007, 352:305-310.

Sellami A, Veenstra JA: SIFamide acts on fruitless neurons to modulate sexual behavior in *Drosophila melanogaster*. *Peptides* 2015, 74:50-56.

We have added these references in the revised manuscript.

Minor edits

1. The sentence on line 25-26 is awkward: "...to understand THEM" - what is them referring to?

We have re-wrote the sentence "...are key steps to understand how neural circuits control behaviors".

2. Line 29 (see also Line 66): P1 neurons a "command center". The P1 neurons are part of a command circuit.

We have added "part of" as suggested.

3. Line 34: Is it really clear that Fru neurons regulate ALL aspects of sexual behaviors. This is a strong statement.

We have changed "all" to "most" to more accurately reflect roles of *fru* neurons.

4. Line 613: what is positively and negatively purified?

This is a mistake. We have corrected the sentence to "The antiserum was purified and used at 1:100 dilution".

Our responses to comments are immediately after each comment, and in blue:

Reviewer #2 (Remarks to the Author):

This paper reports on the discovery of a few fru-positive interneurons (MP1) that inhibit activities in P1 neurons, the primary decision-making center for male courtship behavior. The newly discovered courtship-inhibiting neurons likely release the neuropeptide Drosulfakinin-2 (DSK-2) to suppress activities in P1 neurons, which nonetheless turned out to be presynaptic to the Dsk-positive interneurons. Remarkably, activity-dependent GRASP signals at the P1 – MP1 neuronal connections appeared to be enhanced by male – male social interactions, while other neural inputs impinging on MP1 were differently affected by rearing conditions. Overall, the experiments were carefully conducted using an impressive array of newly-emerged techniques, which well substantiated the conclusion that MP1 represents a state-dependent negative regulator of courtship activities in flies. This work is worth publishing after the clarification of certain ambiguity.

1. The authors emphasize the importance of neural integration of internal states and external stimuli. This raises a question what external stimuli may provoke MP1 activities. Social experience must be converted into some neural codes, likely initiated by sensory cells and ultimately impacts MP1. Although the identification of such an input pathway to MP1 is beyond the scope of this paper, the authors are encouraged to discuss which sensory modality is involved and how the social cue is conveyed to MP1.

We agree with the reviewer that how Dsk neurons integrate internal states and social experiences is of particular interest, but still not very clear. We have added a paragraph in the discussion section to discuss what may activate Dsk neurons and how Dsk may function as an inhibitory signal for sexual behaviors in flies.

2. “FRU[M] is both necessary and sufficient” (line 52) is not warranted and thus to be removed (see more details: Yoshihara and Yoshihara, 2018 <https://www.ncbi.nlm.nih.gov/pubmed/29757057>).

We thank the reviewer for pointing out this and have made textual changes as suggested. We also thank the reviewer for recommending this wonderful paper on the use of “necessary and sufficient” in writing.

3. The authors found a prolonged inhibitory effect after transient activation of MP1 on male courtship (lines 123-129). How could MP1 activation give rise to a persistent effect after the cessation of optogenetic stimulations? What would be the biological advantage of such long-lasting suppression of male courtship? These issues merit discussion.

It is very interesting that after the cessation of optogenetic stimulation of Dsk neurons, courtship inhibition still lasts for minutes. Perhaps after termination of optogenetic activation, those released DSK peptides still function on their receptors and requires some time for degradation. Similarly, P1 activation also promoted a persistent behavioral state that promotes male courtship and aggression. We think these persistent effects may represent arousal states that normally last for some time (e.g. minutes). Although we have no direct evidence, but the impact of internal states (e.g.

(feeding states) and past experiences (e.g. group-housing conditions) all have long-lasting effects on behavior. Future studies on how activation of Dsk and P1 neurons has persistent behavioral effects will deepen our understanding of how such persistent effect was generated. We have added a brief discussion regarding this phenotype in the discussion section.

4. Potential connections from different sets of dsx-positive neurons to MP neurons appear to be enhanced under different social contexts (lines 272-283; Fig. 4j-n). What are the possible neural groups that are responsible for altered activities in ROIs 1-3?

There are much more GRASP signals from dsx neurons to Dsk neurons, compared with signals from P1 to Dsk neurons. We know that dsx neurons in the brain can be divided mainly as P1 (or pC1), pC2 and pCd neurons, and we think ROI-2 is mainly from P1 neurons (as also seen in Fig. 5h), but not sure if GRASP signals in ROI-1 and ROI-3 come from pC2, pCd or other P1 neurons. We will need clean LexA drivers for pC2 and pCd at this stage to discriminate where these signals come from. We tried R41A01-LexA and R40F04-LexA that weakly labels pCd and pC2 neurons as well as many other neurons (Zhou et al., 2014, Neuron), and did not see any GRASP signals from these neurons to Dsk neurons, probably due to their weak expression. We have mentioned these results in the result section, and we believe that better genetic tools will help to answer these questions in the future.

5. The authors claim that immunoreactivity to the anti-DSK antibody at presynaptic terminals is decreased after TrpA1-mediated activation of MP neurons. However, the data shown in Supplementary Fig. 6b indicate that the anti-DSK staining intensity decreased throughout a neuron, not just at presynaptic terminals. In this case, the way to describe the observed changes is misleading. If presynaptic terminals were more sensitive to forced activation than other neural parts, the authors should show higher magnification views of a neuron stained for DSK, making it possible to compare the staining intensity among different portions of the cell.

We thank the reviewer for pointing out the inaccuracy in our description. Our data shows an overall decrease of DSK as shown in Supplementary Fig. 6b. So we edit the text as follows: We also found that activation of Dsk neurons in DskGAL4/UAS-TrpA1 males at 30°C for 30 min decreased DSK immunoreactivity in many parts of Dsk neurons including soma, suggestive of DSK secretion in responsive to neuronal activation.

We also quantified DSK expression in multiple regions as circled in Supplementary Fig. 6a, and found that in all these regions anti-DSK signals decreased after activation of Dsk neurons for 30 min. Supplementary Fig.6 is accordingly updated.

6. Functional GRASP data suggested that P1 synapses on to DSKergic MP1, whereas the DSK receptor CCKLR-17D3 was expressed in P1. In fact, the authors seem to imply bidirectional P1 – MP neuron communications (lines 475-461). What mechanism could mediate the MP-to-P1 information flow? Do you suggest an involvement of systemic DSK or any retrograde actions across synapses? A schematic illustrating hypothetical neural and humoral pathways that may contribute to P1-MP interactions for regulating courtship and sleep would help readers understand the provisional network underlying behavioral choice switch.

Our results suggest a functional loop between Dsk and P1 neurons, and we are also aware that many dsx neurons (not only P1) act on Dsk neurons, and Dsk may further function on CCKLR-17D3 neurons including many fru and dsx neurons. Thus this dsx-Dsk-17D3 circuit is much more complex than a potential P1-Dsk-P1 loop. In other words, this potential P1-Dsk-P1 loop is a part of dsx-Dsk-17D3 circuit. We think dsx neurons act on Dsk neurons through neurotransmitters (e.g. Ach, as found by Zhou et al., 2014 and Chen et al., 2017) with direct synaptic connection, and Dsk neurons act on 17D3 (many fru/dsx neurons) via DSK secretion that does not necessarily via direct synaptic connection. We added an illustration of this pathway as Fig. 7l and made textual changes accordingly.

7. Based on functional GRASP assays, the authors argue that male-to-male interactions potentiate synaptic transmission from P1 to MP neurons (lines 269-272). However, one cannot exclude the possibility that stronger GRASP signals observed at P1-to-MP synapses were a secondary consequence of the general elevation of P1 activity levels upon male-to-male interactions, not reflecting any augmentation specific to these synapses. The authors' observation that many connections between different sets of dsx-positive neurons and MP neurons were boosted by social experience seems to be consistent with this idea. Kohatsu and Yamamoto (2015) previously reported that responsiveness to moving visual targets of putative P1 neurons is susceptible to housing conditions, at least in fru mutant males. Do you think that enhanced P1-to-MP transmission ultimately suppresses male-directed courtship in these males? The discussion needs to be expanded to include possible mechanisms for enhanced GRASP signals and behavioral roles of such enhanced transmission.

We think enhanced GRASP signals from P1-Dsk neurons are not from general elevation of P1 activity. Although P1 activity in response to moving targets could be enhanced by housing conditions in fru mutant males, it was also found in wild-type males (fru+) that group-housing actually decreases excitability of P1 neurons (Inagaki et al., 2014).

Since group-housing enhanced synaptic transmission from P1 neurons to *Dsk* neurons, we asked whether group-housing may affect male courtship in a DSK-dependent manner. So we first tested male-female courtship by single-housed males or group-housed males (11 males, the same as used in GRASP experiments), and found no difference of male courtship in wild-type control males and *Dsk* knocked-down males (see Supplementary Fig. 8a). We then repeated male courtship, but tested in dark condition and used headless females as targets, and found that group-housing significantly reduced male courtship towards headless females in dark in wild-type males; however, such group-housing induced courtship reduction is not seen in *Dsk* knocked-down males, as they courted equally high under two rearing conditions (see Supplementary Fig. 8b). Thus we provided further experimental evidence that male-male group-housing increases DSK expression and reduces male courtship. These results were added as Supplementary Fig. 8, and we also made textually changes accordingly.

Our responses to comments are immediately after each comment, and in blue:

Reviewer #3 (Remarks to the Author):

In this study, Wu and colleagues explore the intriguing question of how internal states interact and are regulated by examining how peptidergic systems modulate *Drosophila* courtship. They demonstrate that activation of a small number of Fru+ drosulfakinin-expressing neurons in the medial protocerebrum strongly suppresses courtship and suggest that this effect appears to be mediated by drosulfakinin acting via one of its cognate receptors to modulate the activity of male-specific P1 neurons that serve to initiate courtship behavior. The authors show that co-activation of these Dsk+ neurons and P1 neurons lead to intermediate phenotypes in both courtship and sleep assays, in accord with an antagonistic interaction between these two populations. Broadly, I found the results interesting and convincing, and appreciate the general thrust of the manuscript. One limitation of this work is that the functional logic of the Dsk modulation is largely derived from behavioral manipulations, many of which rely on artificial activation of Dsk and/or P1 neurons. If the authors could more clearly elucidate the interactions and chain-of-command in the proposed circuitry, this work would certainly represent an advance in thinking about the regulation of motivational states.

One major concern in the current manuscript is that the natural role of Dsk neurons in natural courtship is not clear, nor exactly over what time scales they affect downstream circuitry. While the authors nicely demonstrate that acute activation of these neurons strongly reduce courtship, and that this effect is dependent on one of the Dsk receptors, I remain a little unclear from the data presented about when these neurons would be normally activated to regulate courtship in this manner. This concern is more pronounced by the lack of behavioral evidence that silencing Dsk neurons or knocking out Dsk has an impact on normal courtship behavior. A related point is that the authors should discuss the time scale on which these neurons are proposed to regulate behavior—acutely, over minutes (as Fig. 1 would suggest) in response to changes in sensory input, or over hours or days (as Fig. 4 would suggest) or the lifetime of the fly (as Fig. S8 would suggest) in responses to past experience or changes in their mating drive?

We thank the reviewer for appreciation of the general thrust of our manuscript. We agree with the reviewer that how Dsk neurons were naturally activated to modulate male courtship is still unclear, although we have shown that Dsk neurons receive inputs from dsx neurons including P1 neurons in an experience-dependent manner, and multiple physiological states influence expression of DSK as well as CCKLR-17D3. In fact, how Dsk neurons are regulated by internal states and past experiences to antagonize P1 neuronal function to determine a male's motivational state to court or not, will be one of our main focus in the future. As this requires intensive studies including detailed analysis of Dsk-P1 (or non-P1 but dsx-expressing neurons, e.g. pC2 and pCd) functional loop under different conditions, which exceeds the scope of this study, we instead added a paragraph in the discussion section to discuss how Dsk neurons could be activated and in which time scale they function.

The logic behind P1 neurons synapsing directly onto Dsk neurons should also be clarified. Intuitively, if the authors believe that the Dsk neurons are acting via P1 neurons, it would seem that P1 activation would simultaneously drive and suppress

courtship. This again raises a question regarding the time-scale over which these neurons function, but also raises the possibility that Dsk suppression could be mediated by a distinct downstream target. Indeed, GRASP signals are much greater when all *dsx*⁺ neurons are included suggestive that Dsk signaling may just as readily act to suppress courtship via a common target of both P1 and Dsk⁺ neurons. The authors should experimentally address whether Dsk signaling onto P1 neurons is driving courtship suppression, perhaps by knocking down the CCKLR receptors in the neurons targeted by the split-P1 GAL4 line.

We thank the reviewer for this suggestion. Although we found a potentially functional loop between Dsk and P1 neurons, we are aware that many *dsx* neurons (not only P1) act on Dsk neurons, and Dsk may further function on CCKLR-17D3 neurons including many *fru* and *dsx* neurons. Thus this *dsx*-Dsk-17D3 circuit is much more complex than a potential P1-Dsk-P1 loop. Our future studies will first dissect the function of P1-Dsk-P1 loop in modulation of male courtship, and then expand to *dsx*-Dsk-17D3, as this requires much more genetic tools including cleaner drivers for subsets of *dsx* neurons (e.g. *pC2* drives).

In terms of whether Dsk function onto P1 neurons to suppress male courtship, as we mentioned above on the *dsx*-Dsk-17D3 circuit, we think Dsk may act on many *fru*/*dsx* neurons to suppress male courtship, not only P1 neurons. Nevertheless, we performed CCKLR-17D3 knock down experiments using *fru*GAL4, *dsx*GAL4 and P1-splitGAL4 (only labels ~10 pairs of P1 neurons), and we found that all these manipulations did not affect male courtship. The reason might be that since the courtship index towards virgin females is already very high, and knocking down inhibitory modulators does not further increase courtship. We also tried overexpression of CCKLR-17D3 in all *fru* neurons, *dsx* neurons and P1 neurons, and found that male courtship was significantly reduced when CCKLR-17D3 is overexpressed in all *fru* or *dsx* neurons, but not P1 neurons alone. These results support the notion that Dsk acts on many *fru* and *dsx* neurons including P1, but not only rely on P1's function. These data were now added as Supplementary Fig. 9. We also made textual changes into the revised manuscript accordingly.

The authors rule out the possibility of Dsk neurons driving a competing behavior (which would parsimoniously explain the results) on the basis of five arguments (lines 426-432). It is not fully clear to me, however, that this conclusion logically flows from these arguments nor other data. For example, there appears to be a systemic decrease in the velocity of DskGal4 > dTrpA1 neurons relative to controls (Fig. 3g-j; the statistical test should account for repeated measures). The monomorphic nature of many of these neurons would further strengthen this hypothesis. The authors should discuss the possibility that Dsk neurons drive a competing behavior, phenotypically shunting courtship.

We thank the reviewer for this suggestion and have made textual changes to this argument more accurately. Below is a part of textual changes: As DSK signaling modulates multiple behaviors, one may argue that its role in male courtship is not specific, e.g., activation of *Dsk* neurons may drive a competing behavior that phenotypically shunting male courtship. Although we cannot exclude such possibility, we listed a number of evidences as also summarized in Fig. 8 to support DSK's role with specificity in courtship inhibition.

Lastly, there appears to be a quite strong suppression of courtship in the control empty Gal4 line in Fig. 1d. This is a very good control, but the authors should discuss this effect, and their choice to not use it as a control for the TrpA1 experiments in Fig. 3.

We have added additional explanatory text: Note that the empty *GAL4* control flies also showed reduced courtship under red light, which may be due to genetic background and/or red light perturbation, and we used other control lines (e.g. *DskGAL4/+*) in our later experiments.

REVIEWERS' COMMENTS:

Reviewer #1 (Remarks to the Author):

I have no further comments, although authors evaded some queries

Reviewer #2 (Remarks to the Author):

The authors have adequately addressed all concerns I raised and I am happy to recommend the manuscript for publication.

Reviewer #3 (Remarks to the Author):

In this revised manuscript, the authors have made significant textual improvements and added several new experimental results. The authors now demonstrate that overexpression of the cognate Dsk receptor CCKLR-17D3 in fruM-expressing and dsx-expressing neurons, but not P1 neurons, phenocopies direct activation of Dsk-expressing neurons by decreasing courtship. They moreover show that, in contrast to Dsk neurons, exogenous activation of SIFamide expressing neurons does not suppress male-female courtship. The authors have also added new data showing that RNAi-mediated knock-down of Dsk prevents a group-housing induced suppression in courtship towards headless females. Overall, I continue to be enthusiastic about the paper, although I have a few reservations that should be addressed at least with further textual clarification.

One concern with the current text is that the authors continue to imply connectivity between dsk-expressing neurons and P1 neurons that is not strictly feed-forward, but rather recurrent in nature (e.g. lines 89-91, 394-395, Fig. 8). As I noted in my initial review, I think P1 neurons would provide a parsimonious target via which Dsk neurons could regulate courtship, but such a model would seem to introduce a complicated activity cascade in which courtship is simultaneously promoted and suppressed. The authors have now carried out new CCKLR-17D3 knock-down and overexpression experiments in P1 neurons, and found no effects on courtship. This, along with the lack of direct synaptic connections from Dsk neurons to P1 neurons, offers little support for a model where Dsk neurons directly communicate with P1 neurons. A recurrent loop, as depicted in Fig. 8, is not really recurrent if it is propagating information from and to distinct dsx neural populations.

The physiological role of Dsk neurons in regulating courtship also remains a bit obscure. The authors have added new data showing that knock-down of Dsk prevents a reduction in courtship towards a headless female in the dark following group housing. While this is a useful addition in that it is the first demonstration that there is a behavioral phenotype from perturbing natural Dsk signaling, this behavior appears to be quite contrived. Why is a phenotype only apparent once courtship levels are dampened by "restricted conditions" e.g. introducing a headless female in the dark? Moreover, why would group-housed males benefit from suppressed courtship towards a stationary female in the dark? The authors should discuss the ethological interpretation of this behavior, and the importance of Dsk in its regulation.

Lastly, several of the resolution and saturation of several confocal images in the paper should be improved. The 17D3GAL4 intersection with FruLexA images in Figure 7 appears overly saturated, making it difficult to see finer structures. For example, many of the structures labeled 7h are not visible in 7g (before the intersection), and the finer details of the P1 neurons are difficult to identify (7j-k). The resolution of images in all other figures, except Figure 3, should also be

improved.

Our responses to comments by reviewer 3 are listed below, and in blue:

Reviewer #3 (Remarks to the Author):

In this revised manuscript, the authors have made significant textual improvements and added several new experimental results. The authors now demonstrate that overexpression of the cognate Dsk receptor CCKLR-17D3 in fruM-expressing and dsx-expressing neurons, but not P1 neurons, phenocopies direct activation of Dsk-expressing neurons by decreasing courtship. They moreover show that, in contrast to Dsk neurons, exogenous activation of SIFamide expressing neurons does not suppress male-female courtship. The authors have also added new data showing that RNAi-mediated knock-down of Dsk prevents a group-housing induced suppression in courtship towards headless females. Overall, I continue to be enthusiastic about the paper, although I have a few reservations that should be addressed at least with further textual clarification.

We thank the reviewer for being enthusiastic about the manuscript, and providing very thoughtful comments.

One concern with the current text is that the authors continue to imply connectivity between dsk-expressing neurons and P1 neurons that is not strictly feed-forward, but rather recurrent in nature (e.g. lines 89-91, 394-395, Fig. 8). As I noted in my initial review, I think P1 neurons would provide a parsimonious target via which Dsk neurons could regulate courtship, but such a model would seem to introduce a complicated activity cascade in which courtship is simultaneously promoted and suppressed. The authors have now carried out new CCKLR-17D3 knock-down and overexpression experiments in P1 neurons, and found no effects on courtship. This, along with the lack of direct synaptic connections from Dsk neurons to P1 neurons, offers little support for a model where Dsk neurons directly communicate with P1 neurons. A recurrent loop, as depicted in Fig. 8, is not really recurrent if it is propagating information from and to distinct dsx neural populations.

We appreciate the concern raised by the reviewer that Dsk neurons may not act on P1 neurons, as knocking down CCKLR-17D3 in P1 neurons does not affect male courtship. Although we did not directly prove a functional flow from Dsk to P1 neurons, we found that P1 neurons express DSK receptor CCKLR-17D3, which promoted us to propose a functional loop between Dsk and P1 neurons. Since the lack of direct functional evidence, we thus changed our text accordingly, e.g., not concluding “a functional loop” in the result section, and only propose such possibility in the discussion section and working model (by saying “a potential functional loop”).

The physiological role of Dsk neurons in regulating courtship also remains a bit obscure. The authors have added new data showing that knock-down of Dsk prevents a reduction in courtship towards a headless female in the dark following group housing. While this is a useful addition in that it is the first

demonstration that there is a behavioral phenotype from perturbing natural Dsk signaling, this behavior appears to be quite contrived. Why is a phenotype only apparent once courtship levels are dampened by “restricted conditions” e.g. introducing a headless female in the dark? Moreover, why would group-housed males benefit from suppressed courtship towards a stationary female in the dark? The authors should discuss the ethological interpretation of this behavior, and the importance of Dsk in its regulation.

Male courtship behavior under the regular condition is very robust (mostly >80% percent of time courting), as different sensory modalities are redundant for male courtship (Krstic et al., Sensory Integration Regulating Male Courtship Behavior in *Drosophila*, 2009). The restricted courtship condition we used here allowing only olfactory and gustatory inputs from female targets to stimulate male courtship, thus is more sensitive to detect courtship reduction. We now added this reference to clarify the purpose of using such restricted courtship assay. We also briefly discussed how male-male group-housing experience would reduce later courtship in the discussion section.

Lastly, several of the resolution and saturation of several confocal images in the paper should be improved. The 17D3GAL4 intersection with FruLexA images in Figure 7 appears overly saturated, making it difficult to see finer structures. For example, many of the structures labeled 7h are not visible in 7g (before the intersection), and the finer details of the P1 neurons are difficult to identify (7j-k). The resolution of images in all other figures, except Figure 3, should also be improved.

We thank the reviewer for pointing this out, which may be due to file compression during submission. We have now adjusted all images in the manuscript to the best resolution as we can, and will upload individual uncompressed figures.